# A size and space structured model of tumor growth describes a key role for protumor immune cells in breaking equilibrium states in tumorigenesis

**Kevin Atsou**[1¤], **Fabienne Anjuère**[2], **Véronique M. Braud**[2], **Thierry Goudon**[1]*

**1** Université Côte d'Azur, Inria, CNRS, LJAD, Parc Valrose, Nice, France, **2** Université Côte d'Azur, CNRS, Institut de Pharmacologie Moléculaire et Cellulaire UMR 7275, Valbonne, France

¤ Current address: Inria-Inserm Team COMPO, Inserm U1068, CNRS UMR7258, Institut Paoli-Calmettes Pharmacy faculty, Aix-Marseille University, Marseille, France
* thierry.goudon@inria.fr

**Data Availability Statement:** Relevant data are within the manuscript. Codes and numerical data are available at the URL https://github.com/atsoukevin93/tumorgrowth.

## Abstract

Switching from the healthy stage to the uncontrolled development of tumors relies on complicated mechanisms and the activation of antagonistic immune responses, that can ultimately favor the tumor growth. We introduce here a mathematical model intended to describe the interactions between the immune system and tumors. The model is based on partial differential equations, describing the displacement of immune cells subjected to both diffusion and chemotactic mechanisms, the strength of which is driven by the development of the tumors. The model takes into account the dual nature of the immune response, with the activation of both antitumor and protumor mechanisms. The competition between these antagonistic effects leads to either equilibrium or escape phases, which reproduces features of tumor development observed in experimental and clinical settings. Next, we consider on numerical grounds the efficacy of treatments: the numerical study brings out interesting hints on immunotherapy strategies, concerning the role of the administered dose, the role of the administration time and the interest in combining treatments acting on different aspects of the immune response. Such mathematical model can shed light on the conditions where the tumor can be maintained in a viable state and also provide useful hints for personalized, efficient, therapeutic strategies, boosting the antitumor immune response, and reducing the protumor actions.

## Introduction

The immune system can both constrain and promote tumor development through several complex processes, encompassed in the concept of cancer immunoediting [1]. The antagonistic effects of the immune response on tumor growth shape the different phases that have been identified to characterize their interaction: elimination, when the immune system is able to detect and eradicate the developing tumors; equilibrium, when the immune system is able to

**Funding:** This work was supported by the French Government (National Research Agency, ANR) through the "Investments for the Future" programs LABEX SIGNALIFE ANR-11- LABX-0028 and IDEX UCAJedi ANR-15-IDEX-01. The funders had no role in study design, data collection and analysis, decision to publish, or preparation of the manuscript.

**Competing interests:** The authors have declared that no competing interests exist.

maintain the tumor expansion in a cancer-persistent state; and escape, when the tumor develops in an uncontrolled manner [1, 2].

In this context, the identification of the immune components of the tumor microenvironment (TME) reveals valuable information about the stage of cancer development and helps predict patient outcome. This concept called "the immune contexture" has improved the classification of cancers [3, 4]. The antitumor immune response is characterized by the activation and the recruitment of innate immune cells such as natural killer (NK) cells, tumor-associated neutrophils (TAN-N1), tumor-associated macrophages (TAM-M1) and adaptive immune CD8$^+$ T cells. They migrate to the tumor site where they can eliminate tumor cells. They have been found to be highly active on early-stage tumors and associated with good clinical outcome [5, 6]. While this antitumor immune response can be expected to control tumor growth or maintain their development in a viable equilibrium, later phases are characterized by an uncontrolled tumor growth associated with a shift of the immune response towards protumor functions and the establishment of multiple mechanisms of immunosuppression [7]. Among others, the ratio of effector immune cells/protumor immune cells is considered as a relevant indicator of patient survival, the higher the ratio, the better the patient vital prognostic [8]. The ratio evolves dynamically: tumor cells and other components in the TME can produce inhibitory factors such as anti-inflammatory cytokines, interleukins 10 and 4 (IL-10 and IL-4), Transforming Growth Factor-beta (TGF-$\beta$) which favor the polarization of antitumor immune cells into protumor ones. For instance, antitumor neutrophils and macrophages are converted into protumor TAN-N2 and TAM-M2 [9, 10]. They are part of a pool of myeloid-derived suppressor cells (MDSCs) which can also be directly recruited from the bone marrow [11]. They promote tumor growth, tissue remodeling, angiogenesis and suppress adaptive immunity [12]. Moreover, the antigen-presenting cells such as dendritic cells (DC) become tolerogenic which leads to exhausted and tolerant T cells, apoptosis of T cells and to the priming and proliferation of regulatory T cells (Tregs) [12, 13]. Besides, tumors subvert mechanisms of immune self-tolerance and inhibit antitumor immune responses through the use of immune checkpoints. The TME is therefore infiltrated with hyporesponsive exhausted T cells [14]. Blockade of these immune checkpoints as exemplified with anti-CTLA4 and anti-PD-1/PD-L1 restore efficient effector functions and has revolutionized cancer treatment [15].

Mathematical modeling might shed some light on these complex interactions, and, based on numerical simulations, provide useful information to elaborate more efficient therapeutic strategies. Quite intricate ordinary differential systems have been developed so far, see for instance [16–22]. Further references and discussion of the various viewpoints can be found in the review [23]. In [24], we proposed a system of partial differential equations (PDE), describing the earliest stages of the tumor/immune system interactions. The system couples an integro-differential equation for the size-structured population of tumor cells, inspired from [25–28], to a convection-diffusion equation for the space-structured immune cells. The latter accounts for chemotaxis mechanisms that drive the immune cells towards the tumor. This model, which only considers the antitumor actions of the immune system, is able to reproduce the equilibrium phase: the large time behavior of this PDEs system is a state where residual tumor cells and a positive concentration of active immune cells exist in equilibrium. However, the simple model of [24] does not consider the contribution of immune cells with protumor functions and the establishment of numerous mechanisms of immunosuppression. This is the issue addressed in this work. To be more specific, our purpose is two-fold:

- First, we incorporate in the model protumor effects that can both reduce the antitumor immune response and strengthen the factors of tumor growth. We shall see that the

protumor immune response can break the equilibrium and lead to an escape phase characterized by the uncontrolled growth of the tumor.

- Second, we complete the model by discussing the effect of different type of targeted treatments that can act on the immune response, either by restoring the effector functions of antitumor cells which became exhausted as a result of chronic activation and protumor factors, or by limiting the recruitment of protumor immune cells. The investigation demonstrates the interest of combining both approaches.

The paper is organized as follows. We introduce the modeling assumptions and we set up the equations in the **Mathematical Model** section. We pay a specific attention to the description of the activation of protumor mechanisms, based on the action of cytokines, in response to the growth of the tumor mass. The modeling assumptions naturally induce some delay mechanisms in the protumor response. We also bring out some mathematical properties of the model. We start by considering a simplified situation which reduces the model to a nonlinear system of ODEs. We identify several stationary solutions, free of tumors, free of protumor cells or with all populations of immune cells, and we discuss their linear stability. This discussion provides some hints on the role of the parameters. Next, we establish the existence of equilibrium states for the full size- and space-dependent problem, extending to the model with protumor activities the results of [24]. The **Results** section is devoted to the numerical investigation of the PDE system. We show that depending on the parameters of the model, the solutions either converge to an equilibrium or describe an escape phase with an unlimited growth of the tumor. These results illustrate the critical role of the protumor immune responses. Next, we address specifically the question of treatments. As detailed below, the immune response might have multi-faceted protumor actions. Among other, effector immune cells can become exhausted, a state where they are hyporesponsive and cannot kill the tumor. We consider the effect of treatments that either restore antitumor activity or reduce the recruitment of protumor immune cells. The investigations demonstrates the interest of combining both approaches and discuss the role of the dose and time of administration.

## Mathematical model

A schematic overview of the geometry and the leading mechanisms that guide the construction of the mathematical model is provided in Figs 1 and 2.

### Modeling assumptions

We take into account three populations of interacting cells:

- the cytotoxic effector cells including CD8$^+$ T cells and NK cells as well as myeloid effector cells, TAN-N1 and TAM-M1 that will be referred to as the "antitumor" immune cells;

- the "protumor" immune cells, including Treg, MDSCs, TAN-N2 and TAM-M2 favoring tumor growth;

- the tumor cells.

For the purpose of the mathematical modeling, we are collapsing into single "averaged" quantities the behavior of several distinct cells, that might have different developmental properties and interactions mechanism, to focus on the outcome of their action on tumor growth. The construction of the model uses the same basis as in [24], to which we incorporate the "protumor" immune cells. The modeling assumptions are as follows.

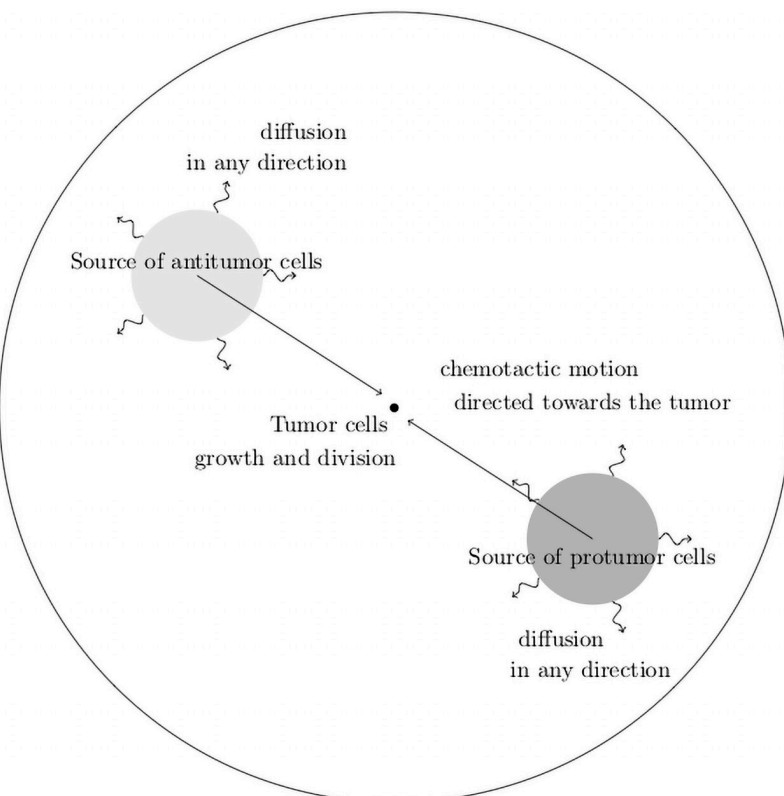

**Fig 1. Schematic view of the geometry of the mathematical model.**

A.1. the environmental constraints such as nutrient concentrations, temperature, etc. are assumed to be constant. Nevertheless, in late stages of tumor growth, some phenomena such as hypoxia or difficulties in accessing the nutrients can limit the tumor cell expansion;

A.2. the state of the tumor cells is characterized by their size (with a similar setting, it could be their content of cyclins as well [25, 29]); the dynamics of the tumor cells is driven by two phenomena: each tumor cell grows with a certain rate, possibly depending on its size, and it can divide into daughter cells;

A.3. activated antitumor immune cells are able to destroy the targeted tumor cells;

A.4. activated protumor immune cells suppress the antitumor immune cells by direct contact or by the release of soluble substances (like immunosuppressive cytokines);

A.5. activated protumor immune cells favor the tumor growth by enhancing the growth rate of the tumor cells and by favoring angiogenesis.

Moreover, the tumor cells produce several signals of chemical nature (cytokines and chemokines), which drive the immune response as follows:

A.6. a chemotactic signal, proportional to the tumor mass, induces a potential, the gradient of which drives the anti and protumor cells towards the TME;

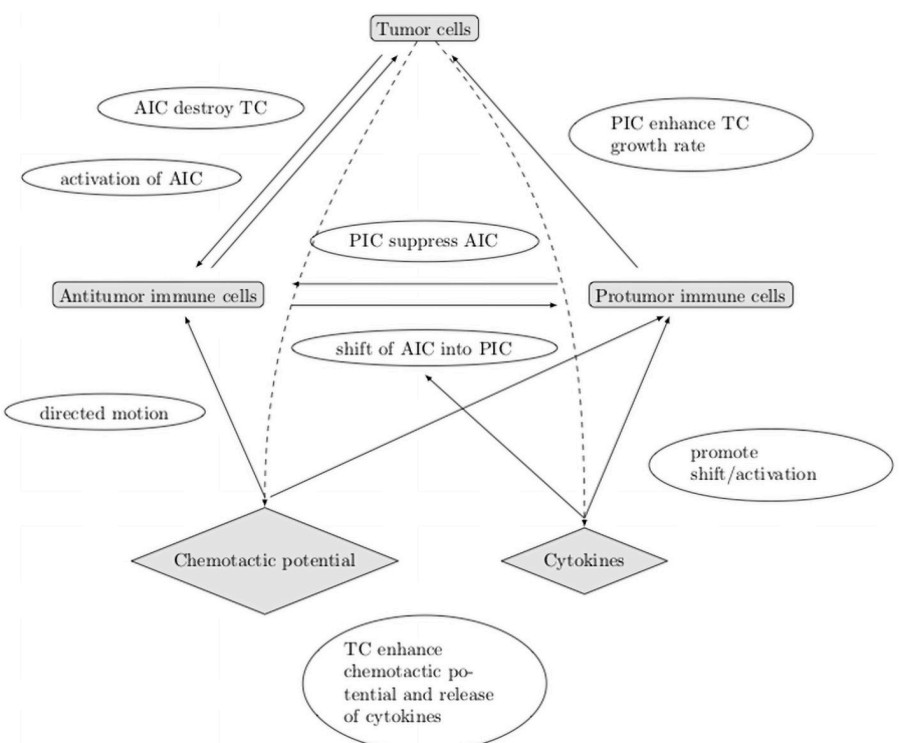

**Fig 2. Schematic view of the leading mechanisms that guide the construction of the mathematical model.** AIC: antitumor cells, PIC: protumor cells, TC: tumor cells.

A.7. the tumor antigen-specific CD8⁺ T cells are activated by APCs in draining lymph nodes and recruited to the tumor site. The NK cells as well as the myeloid cells (TAN-N1, TAM-M1) are recruited from the circulation and activated from a bath of non-activated immune cells at the tumor site. The signal that defines the recruitment/activation rate is directly related to the tumor mass.

A.8. similarly, protumor immune cells can be recruited from a bath of immune cells (that might differ from the source of antitumor immune cells, though) according to a signal directly related to the tumor mass;

A.9. the signal triggers the shift of certain antitumor immune cells into protumor immune cells.

Assumptions **A.1**–**A.3** appeared in [24] where they are discussed in details. The protumor effects become sensitive in a later stage of the tumor growth, and, as we shall see, play a central role in the transition to the escape phase. Assuming a constant growth rate of the tumor cells becomes questionable in this regime, and we shall model it by means of a Gompertz law, which accounts for size-limitation mechanisms, see (1) below. Assumption **A.4** describes immunosuppression mechanisms mediated by protumor immune cells. In addition to the contact-dependent suppression of antitumor immune cells, the secretion of immunosuppressive cytokines abrogates the effector functions of T cells and NK cells and negatively modifies their proliferation. It triggers the reverse conditioning of DCs and can induce the apoptosis of effector T cells through the depletion of IL-2 from the TME. It is worth bearing in mind that not all the antitumor immune cells are eliminated: they become exhausted and can no longer

kill tumor cells; however, they can be reactivated by the action of some treatment (i. e. anti-PD-1 therapy). Assumption **A.5** corresponds to the overexpression of VEGF by protumor immune cells favoring the accumulation of microvessels supplying the tumor in nutrients. Assumption **A.6** already appeared in [24]; here the chemotactic mechanisms apply on both type of immune cells. Assumption **A.7** corresponds to a rough description of the complex activation and recruitment mechanisms which are related to the tumor mass.

Similarly, assumption **A.8** corresponds to the recruitment of MDSC coming from the bone marrow and Tregs from the circulation. Assumption **A.9** corresponds to the possible conversion of some immune cells that eliminate the tumor into protumor immune cells (i. e. macrophages and neutrophils becoming TAN M2 and TAM N2, conventional T cells becoming immunosuppressive Treg).

## Construction of the model

Following [24], the model uses two distinct length scales:

- the length scale of the displacement of the immune cells,

- the length scale describing the size of the tumor cells.

The modeling assumes that the former is "infinitely large" compared to the latter. This is motivated by the fact that immune cell displacement (for instance from the lymph nodes toward the tumor site) occurs on several centimeters while the estimated radius of the tumor cells is about a few micrometers. The interactions between the tumor and the immune system are described by the dynamics of the following unknowns:

- *Tumor cell density*. The population of tumor cells is described by $(t, z) \mapsto n(t, z)$, the volumic density of tumor cells. Given $z_2 > z_1 > 0$, the integral $\int_{z_1}^{z_2} n(t, z) \, dz$ gives the number of tumor cells having a size in the interval $[z_1, z_2]$ at time $t$.

- *Cytotoxic effector cell concentration*. The concentration of antitumor immune cells that are actively fighting against the tumor is $(t, x) \mapsto c(t, x)$.

- *Protumor immune cell concentration*. Similarly, $(t, x) \mapsto c_r(t, x)$ stands for the concentration of the protumor immune cells favoring tumor growth.

- *Chemoattractant potential*. We denote by $(t, x) \mapsto \phi(t, x)$ the concentration of the chemical signal (chemokines) that attracts the immune cells towards the TME.

- *Cytokine concentration*. Finally, let $t \mapsto I(t)$ be the concentration of cytokines in the overall TME.

The dynamics of the population of tumor cells is governed by volume growth and cellular division, see [25–28]. We add to these effects a death rate induced by the activated antitumor immune cells. Let $z \mapsto V(z) \geq 0$ stand for the tumor cell growth rate. We can assume it is a positive constant, but in the present context it is more appropriate to adopt a size dependent model, that incorporates size-limitation effects, like the Gompertz law

$$V(z) = rz \ln (b/z), \tag{1}$$

with $r > 0$ and $b > 0$, the maximal size. Accordingly, when using (1), the size variable $z$ lies in the interval $[0, b]$. We refer the reader to [30–34] for derivation and use of this law in tumor growth modeling, in particular when taking into account the limited access to nutrients or necrotic mechanisms, see **A.1**, **A.2**. The simpler case where $V$ is constant can be useful since it leads to explicit formulae that can be used to check more easily whether or not the conjectured

behaviors hold. The cell division mechanism is described by the operator

$$Q(n)(t, z) = -a(z)n(t, z) + \int_z^b a(z')k(z|z')n(t, z') \, \mathrm{d}z', \tag{2}$$

where $a(z')$ stands for the frequency of the division of cells having size $z'$, and $k(z|z')$ gives the size-distribution produced from the division of a tumor cell with size $z'$. Again a simplified framework assumes that these coefficients are constant, but it is likely relevant to make use of more intricate laws, for instance prohibiting any division below a certain threshold. What is crucial for modeling purposes is the requirement

$$\int_0^z z'k(z'|z) \, \mathrm{d}z' = z,$$

which is related to the principle of mass conservation. Indeed, it implies that cell-division does not change the total mass

$$\int_0^b zQ(n) \, \mathrm{d}z = 0.$$

However, the total number of cells increases since $\int_0^b n(t, z) \, \mathrm{d}z \geq 0$. A relevant example is provided by the binary division operator

$$Q(n)(t, z) = 4a(2z)n(t, 2z) - a(z)n(t, z), \tag{3}$$

which describes the situation where cells with size $2z$ split into two daughter cells, both with size $z$. Further relevant examples of division kernels can be found in [35]. The equation will be completed by the boundary condition $n(t, 0) = 0$, which means that there is no production of cells with size 0. For further purposes, let us introduce the following quantities

$$\text{total number of tumor cells}: \mu_0(t) = \int_0^b n(t, z) \, \mathrm{d}z, \tag{4}$$

$$\text{total mass of tumor cells}: \mu_1(t) = \int_0^b zn(t, z) \, \mathrm{d}z. \tag{5}$$

The displacement of the anti- and protumor immune cells is driven by convection and diffusion, over a domain $\Omega \subset \mathbb{R}^N$. For the sake of simplicity we assume they have the same diffusion coefficient $D$ and, here, we shall work with a constant coefficient. The convection is defined by the chemotactic potential $\phi$, which depends on the total mass of the tumor. It obeys the diffusion equation

$$-\nabla_x \cdot (\mathcal{K}\nabla_x \phi) = f(\mu_1)\sigma, \qquad \mathcal{K}\nabla_x \phi \cdot v|_{\partial\Omega} = 0 \tag{6}$$

endowed with the homogeneous Neumann boundary condition. In (6), $x \mapsto \sigma(x)$ is a given form function with zero-mean, $\mathcal{K} > 0$ is a positive coefficient. The coefficient $\mathcal{K}$ could be matrix-valued as well, taking into account further details of the vasculature or the properties of the tissues neighboring the tumor, that govern the supply of immune cells. The strength of the potential depends on the total mass of the tumor through the function $\mu_1 \mapsto f(\mu_1) \geq 0$. It is natural to assume that $f(0) = 0$ and $f$ is non decreasing. A typical example of $f$ is the following

Michealis-Menten functional response:

$$f(\mu_1) = \frac{\mu_1}{\eta + \mu_1}, \qquad \eta > 0. \tag{7}$$

We suppose that $c$ and $c_r$ have the same chemotactic sensitivity $\chi > 0$, and they both satisfy homogeneous Dirichlet boundary condition on $\partial\Omega$: the immune cells far from the tumor are non-activated.

Let us now describe the zeroth order terms of the equations, that differ depending on the considered type of cells. Both type of immune cells is subjected to a death rate $\gamma, \gamma_r > 0$. The antitumor immune cells are recruited from a source of naive immune cells $(t, x) \mapsto S(t, x)$. The activation process is described through a rate $\mu_1 \mapsto g(\mu_1) \geq 0$, which depends on the total mass of the tumor. Again, it is natural to assume $g(0) = 0$ and $g$ is non decreasing. There are two other mechanisms that lead to a loss of antitumor immune cells. First, according to assumption **A.4**, the protumor immune cells suppress effector cells; this is traduced by a loss term

$$-k_c cc_r$$

where $k_c > 0$ is the rate of this reaction. Second, according to assumption **A.9**, certain activated effector cells can be converted into protumor immune cells under the action of cytokines in the TME. This leads to the loss term

$$-k_r I\theta c$$

where $k_r > 0$ is the rate of this conversion, and $x \mapsto \theta(x)$ is a given form function, say a peaked Gaussian, indicating that such processes occur in the vicinity of the tumor. This loss of antitumor immune cells contributes to the gain term for the population of protumor immune cells. Cytokines also activate protumor immune cells from a distant source denoted by $S_r$ according to assumption **A.8**.

The effector cells release cytotoxic substances in the TME. This effect is described by the death term

$$m(c, n)(t, z) = n(t, z) \times \int_\Omega \delta(y)c(t, y) \, \mathrm{d}y \tag{8}$$

in the tumor growth equation. It involves a non negative space-dependent weight $x \mapsto \delta(x)$, which incorporates both the strength of the immune response and a radius of interaction. According to assumption **A.5**, recruited protumor immune cells favor the tumor growth. Therefore the growth rate of the tumor cells is enhanced by the presence of protumor immune cells and it becomes

$$V(z)\left(1 + \int_\Omega b_1(y)c_r(t, y) \, \mathrm{d}y\right) \tag{9}$$

with a certain non negative, radially symmetric and compactly supported kernel $b_1$.

Finally, we turn to the dynamics of the tumor-secreted cytokines, which promote the protumor reactions. The production of such cytokines occurs beyond a certain critical mass, denoted by $m$. Moreover, the cytokine concentration is naturally damped with a constant rate $\tau > 0$. This leads to the ODE

$$\partial_t I = \psi(\mu_1) - \tau I \tag{10}$$

where $\psi$ is a threshold function, non negative and non decreasing. For instance, given a

constant $\bar{\psi} > 0$, it can be defined by:

$$\psi(\mu_1) = \bar{\psi} \begin{cases} (\mu_1 - m), & \mu_1 > m \\ 0, & \mu_1 \leq m \end{cases} \tag{11}$$

We shall need the technical assumptions $\psi(0) = \psi'(0) = 0$, which clearly holds when $m > 0$. The threshold $m$ can be used to describe the degree of inflammation of the tumor environment: the smaller $m$, the more inflamed the environment, and a reduced $m$ can correspond to altered soil, due to systemic effects caused by a primary tumor. Eventually, we arrive at the following system describing the interactions between the tumor cells, the effector and protumor immune cells:

$$\partial_t n + \partial_z \left( V(z) \left( 1 + \int_\Omega b_1(y) c_r(t, y) \, \mathrm{d}y \right) n \right) = Q(n) - m(n, c), \tag{12a}$$

$$\partial_t c + \nabla_x \cdot (c\chi \nabla_x \phi - D\nabla_x c) = g(\mu_1)S - \gamma c - k_r I\theta c - k_c c c_r, \tag{12b}$$

$$\partial_t c_r + \nabla_x \cdot (c_r \chi \nabla_x \phi - D\nabla_x c_r) = I(S_r + k_r \theta c) - \gamma_r c_r, \tag{12c}$$

$$\partial_t I = \psi(\mu_1) - \tau I, \tag{12d}$$

$$-\nabla_x \cdot (\mathcal{K}\nabla_x \phi) = f(\mu_1)\sigma, \tag{12e}$$

$$n(t, 0) = 0, \ c|_{\partial\Omega} = 0, \ c_r|_{\partial\Omega} = 0, \ \mathcal{K}\nabla_x \phi \cdot v(\cdot)|_{\partial\Omega} = 0, \tag{12f}$$

$$n(t = 0, z) = n_0(z), \ c(t = 0, x) = c_0(x), \ c(t = 0, x) = c_r^0(x), \ I(t = 0) = I_0. \tag{12g}$$

We remind the reader that the cell division operator $Q(n)$ and the immune cell-tumor interaction term $m(c, n)$ are defined in (2) and (8) respectively. Table 1 recapitulates the biological meaning of the parameters of the model. We refer the reader to [24] for details on the units and typical values of these quantities; further parameter identification from experimental data and sensitivity analysis is detailed in [36].

## A simplified model to assess the role of the parameters on damping and escape

In order to shed some light on the possible behavior of the solutions and on the role of the parameters, we temporarily adopt a set of simplifying assumptions. Let us consider the very specific case where

- $V$ and $a$ are constant,

- the source $S$ of immune cell is constant,

- the source $S_r = 0$ vanishes and the other parameters $D$, $\mathcal{K}$ are constant and positive,

- the coupling function is linear: $g(\mu_1) = \mu_1$,

- the space variation of the concentrations of immune cells is neglected,

- we consider the binary division model (3) with a constant frequency $a$.

**Table 1. Recap of the main definitions and notations.**

| variable | description |
|---|---|
| $z$ | volume of tumor cells |
| $t$ | time variable |
| $x$ | space variable |
| $n$ | size-density of tumor cells with a volume $z$ |
| $V$ | tumor cells growth rate |
| $a$ | tumor cells division rate |
| $\mu_0$ | total number of tumor cells |
| $\mu_1$ | total volume of tumor cells |
| $c$ | concentration of antitumor cells |
| $c_r$ | concentration of protumor cells |
| $\chi$ | chemotactic coefficient |
| $\phi$ | chemotactic potential |
| $D$ | diffusion coefficient of the immune cells |
| $S$ | source of antitumor immune cells |
| $S_r$ | source of protumor immune cells |
| $\mathcal{K}$ | diffusion coefficient of the chemotactic signal |
| $\sigma$ | chemotactic signal |
| $I$ | cytokine concentration |
| $\gamma$ | death rate of the antitumor immune cells |
| $\gamma_r$ | death rate of the protumor immune cells |
| $\tau$ | damping rate of the cytokine concentration |
| $k_c$ | suppression rate of antitumor cells by the protumor cells |
| $k_r$ | conversion rate of antitumor cells into protumor cells |
| $\theta$ | form function of the conversion of antitumor cells into protumor cells |
| $\delta$ | form function of the antitumor action in the TME |
| $b_1$ | tumor growth rate induced by the protumor cells |
| $m$ | threshold on the tumor volume driving the cytokine activation |

These assumptions, that are used only in this Section, completely disregard the space dependence of the unknowns and certainly lack of biological relevance; the ambition of the discussion of this simplified framework is to provide an intuition on the role of the parameters. In this simple situation, the dynamics is described by the following system of ordinary differential equations for $\mu_0$, $\mu_1$, given by (4) and (5), and the time-dependent concentrations of immune cells and of cytokines:

$$
\begin{cases}
\dfrac{\mathrm{d}}{\mathrm{d}t}\mu_0 = \mu_0(a - \delta c), \\[2mm]
\dfrac{\mathrm{d}}{\mathrm{d}t}\mu_1 = V(1 + b_1 c_r)\mu_0 - \delta\mu_1 c, \\[2mm]
\dfrac{\mathrm{d}}{\mathrm{d}t}c = \mu_1 S - \gamma c - k_r cI - k_c cc_r, \\[2mm]
\dfrac{\mathrm{d}}{\mathrm{d}t}c_r = k_r cI - \gamma_r c_r, \\[2mm]
\dfrac{\mathrm{d}}{\mathrm{d}t}I = \psi(\mu_1) - \tau I.
\end{cases}
\tag{13}
$$

The state

$$
\begin{pmatrix} \mu_0^H \\ \mu_1^H \\ c^H \\ c_r^H \\ I^H \end{pmatrix} = \begin{pmatrix} 0 \\ 0 \\ 0 \\ 0 \\ 0 \end{pmatrix} \tag{H}
$$

is a trivial equilibrium solution to (13) which corresponds to a healthy state. However, we can also find equilibrium states with residual tumor cells, effector immune cells and even protumor immune cells. Indeed, let

$$
\begin{pmatrix} \mu_0^{NP} \\ \mu_1^{NP} \\ c^{NP} \\ c_r^{NP} \\ I^{NP} \end{pmatrix} = \begin{pmatrix} \dfrac{\gamma a^2}{\delta VS} \\ \dfrac{\gamma a}{\delta S} \\ \dfrac{a}{\delta} \\ 0 \\ 0 \end{pmatrix}. \tag{NP}
$$

If $\mu_1^{NP} \leq m$, the threshold for the activation of cytokines, this defines an equilibrium solution with a residual tumor and free of protumor immune cells. Next, let us introduce

$$
Q = \frac{k_r a}{\delta \tau},
$$

and let

$$
\mu_1^P = \frac{\dfrac{\gamma a}{\delta} - Qm - \dfrac{k_c a}{\gamma_r \delta} Qm}{S - Q - \dfrac{k_c a}{\gamma_r \delta} Q}. \tag{14}
$$

be the tumor mass at equilibrium with the presence of protumor immune cells. Indeed, if $\mu_1^P > m$ another equilibrium solution is given by

$$
\begin{pmatrix} \mu_0^P \\ \mu_1^P \\ c^P \\ c_r^P \\ I^P \end{pmatrix} = \begin{pmatrix} \dfrac{a}{V(1 + b_1 c_r^P)} \mu_1^P \\ \mu_1^P \\ \dfrac{a}{\delta} \\ Q \dfrac{\psi(\mu_1^P)}{\gamma_r} \\ \psi(\mu_1^P)/\tau \end{pmatrix} \tag{P}
$$

That the definition of this unhealthy state makes sense requires that the right hand side in (14) is positive. It means that

$$\text{either } S > Q\left(1 + \frac{k_c a}{\gamma_r \delta}\right) > 0 \text{ and } \frac{\gamma a}{\delta} > Qm\left(1 + \frac{k_c a}{\gamma_r \delta}\right)$$

$$\text{or } S < Q\left(1 + \frac{k_c a}{\gamma_r \delta}\right) \text{ and } \frac{\gamma a}{\delta} < Qm\left(1 + \frac{k_c a}{\gamma_r \delta}\right),$$

(15)

imposing constraints on the parameters. Let us discuss the possibility of obtaining the different equilibrium states $\mu_1^{NP}$, $\mu_1^{P}$, depending on the ratio $\frac{a}{\delta}$ between the tumor division rate $a$ and the strength of the immune response $\delta$. It measures the competitiveness between the tumor and the antitumor immune cells. We thus study respectively the sign of $\mu_1^{NP} - m$ and $\mu_1^{P} - m$. On the one hand,

$$\mu_1^{NP} - m < 0 \text{ if and only if } \frac{a}{\delta} < \frac{mS}{\gamma},$$

(16)

and on the other hand

$$\mu_1^{P} - m = \frac{\gamma \frac{a}{\delta} - mS}{S - Q_1 \frac{a}{\delta} - Q_2 \left(\frac{a}{\delta}\right)^2},$$

(17)

where

$$Q_1 = \frac{k_r}{\tau} \quad \text{and} \quad Q_2 = \frac{k_c k_r}{\gamma_r \tau}.$$

Let us denote by

$$X_2 = \frac{-\frac{Q_1}{Q_2} + \sqrt{\left(\frac{Q_1}{Q_2}\right)^2 + \frac{4S}{Q_2}}}{2},$$

(18)

the non-negative root of the denominator in (17). By analyzing the sign of (17), we distinguish two cases

- if $m < \frac{\gamma X_2}{S}$ (relatively small critical mass),

$$\mu_1^{P} - m > 0 \text{ if and only if } \frac{a}{\delta} \in \left(\frac{mS}{\gamma}, X_2\right)$$

(19)

and

$$\mu_1^{P} - m < 0 \text{ if and only if } \frac{a}{\delta} \in \left(0, \frac{mS}{\gamma}\right) \cup (X_2, +\infty).$$

(20)

This is summarized in the following table

| $\frac{a}{\delta}$ | 0 | | $\frac{mS}{\gamma}$ | $X_2$ | | $+\infty$ |
|---|---|---|---|---|---|---|
| $\mu_1^P - m$ | $< 0$ | | $0$ | $> 0$ | | $< 0$ |
| | | | $\mu_1^P$ admissible | | | |
| $\mu_1^{NP} - m$ | $< 0$ | | $0$ | $> 0$ | | $> 0$ |
| | | $\mu_1^{NP}$ admissible | | | | |

There is no admissible equilibrium when $\frac{a}{\delta} \in [X_2, +\infty)$: the aggressiveness of the tumor is strong and the tumor mass certainly blows up.

- if $m > \frac{\gamma X_2}{S}$, we have

| $\frac{a}{\delta}$ | 0 | $X_2$ | | $\frac{mS}{\gamma}$ | | $+\infty$ |
|---|---|---|---|---|---|---|
| $\mu_1^P - m$ | $< 0$ | | $> 0$ | $0$ | | $< 0$ |
| | | | $\mu_1^P$ admissible | | | |
| $\mu_1^{NP} - m$ | $< 0$ | | $< 0$ | $0$ | | $> 0$ |
| | | $\mu_1^{NP}$ admissible | | | | |

Again, there is no admissible equilibrium when $a/\delta$ becomes large.

**Discussion on the stability of the equilibrium points.** The Jacobian matrix evaluated at the healthy state (H) reads

$$J^H = \begin{pmatrix} a & 0 & 0 & 0 & 0 \\ V & 0 & 0 & 0 & 0 \\ 0 & S & -\gamma & 0 & 0 \\ 0 & 0 & 0 & -\gamma_r & 0 \\ 0 & 0 & 0 & 0 & -\tau \end{pmatrix}$$

Since $a > 0$, $J^H$ has a positive eigenvalue and the healthy state is linearly unstable. The equilibrium state (NP), which is free of protumor immune cells, corresponds to the unhealthy state in [24]. The Jacobian matrix at this state reads

$$J^{NP} = \begin{pmatrix} 0 & 0 & -\dfrac{\gamma a^2}{VS} & 0 & 0 \\ V & -a & -\dfrac{\gamma a}{S} & 0 & 0 \\ 0 & S & -\gamma & -k_c\dfrac{a}{\delta} & -k_r\dfrac{a}{\delta} \\ 0 & 0 & 0 & -\gamma_r & k_r\dfrac{a}{\delta} \\ 0 & 0 & 0 & 0 & -\tau \end{pmatrix}$$

and its characteristic polynomial is

$$p(\lambda) = -(\gamma_r + \lambda)(\tau + \lambda)(\lambda^3 + (\gamma + a)\lambda^2 + 2a\gamma\lambda + \gamma a^2).$$

As in [24], we distinguish two cases, which depends on the ratio $\frac{\gamma}{a}$. The ratio compares the death rate of the antitumor immune cells to the tumor cells division rate. We get

- if $\gamma > 4a$, the eigenvalues of $J^{NP}$ are given by

$$\lambda_1 = -\gamma_r, \quad \lambda_2 = -\tau, \quad \lambda_3 = -a, \quad \lambda_4 = \frac{1}{2p}\left(-\sqrt{\gamma(\gamma - 4a)} - \gamma\right),$$

$$\lambda_5 = \frac{1}{2}\left(\sqrt{\gamma(\gamma - 4a)} - \gamma\right).$$

They are all real and negative.

- if $\gamma < 4a$, the eigenvalues are given by

$$\lambda_1 = -\gamma_r, \quad \lambda_2 = -\tau, \quad \lambda_3 = -a, \quad \lambda_4 = \frac{1}{2}\left(-i\sqrt{\gamma(\gamma - 4a)} - \gamma\right),$$

$$\lambda_5 = \frac{1}{2}\left(i\sqrt{\gamma(\gamma - 4a)} - \gamma\right).$$

They all have a negative real part.

Therefore, when admissible ($\mu_1^{NP} < m$), the unhealthy state with no protumor immune cells is always linearly stable. In addition, the ratio $\gamma/a$ discriminates between a damped behavior, and an oscillatory behavior. As observed in [24], the greater the cell division, the faster the oscillations of the tumor mass $\mu_1$. For the equilibrium with protumor cells, the Jacobian matrix becomes full and we are not able to find such explicit formula for the eigenvalues.

We observe on numerical grounds that in the case where the equilibrium states (NP) and (P) coexist ($X_2 < \frac{a}{\delta} < \frac{mS}{\gamma}$), there is either the formation of an equilibrium free of protumor cells, namely as in (NP), or the blow up of the tumor mass due to the activation of protumor immune cells; when $\frac{a}{\delta} > \frac{mS}{\gamma}$, the tumor mass always blows up. Typical results are depicted in Fig 3: when the control occurs, the concentration of antitumor immune cells tends to the equilibrium value $a/\delta$ (Fig 3a)); otherwise, it reaches a constant state below the equilibrium value (Fig 3b–3d). The simulations also show that the control of the tumor is very sensitive to the strength of the source of naive immune cells and to the division rate of the tumor cells. It is worth remarking that the damping of the tumor mass towards an equilibrium can be restored by strengthening the activation law of effector immune cells for large tumor masses, for instance by using $g(\mu_1) = \mu_1^2$, see Fig 4.

## Existence of equilibrium phases for the PDE model

We now go back to the full PDE model (12a)–(12g) and we explain why an equilibrium can be expected in certain circumstances. The analysis and simulations carried out in [24] for the model without protumor immune cells reveal the existence and stability of a cancer-persistent equilibrium. It turns out that the equilibrium phase corresponds to the situation where the

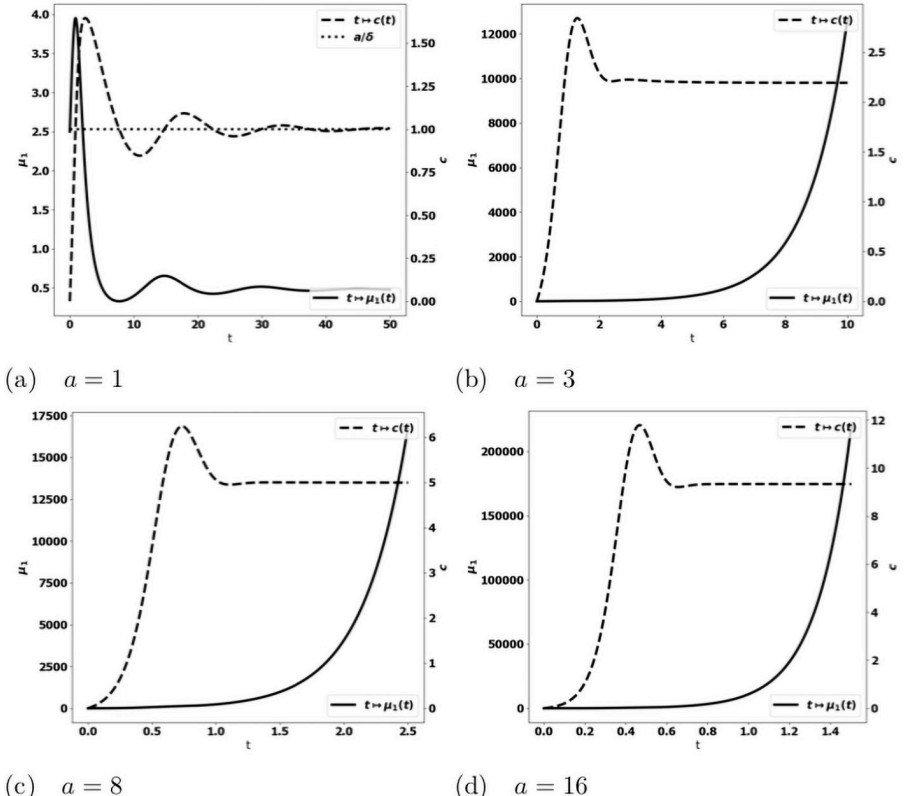

(a)  $a = 1$

(b)  $a = 3$

(c)  $a = 8$

(d)  $a = 16$

**Fig 3. Typical behavior of the solutions of the simplified ODE system (13).** Data: $V = 0.616$, $\delta = 1.$, $S = 1.5$, $k_r = 1.25$, $k_c = 0.1$, $m = 2$. x-axis: time, y-axis: $\mu_1$, mass of the tumor (plain, left axis), and $c$, strength of the active immune cells (dashed, right axis); the expected equilibrium value is $a/\delta$ (dotted line in fig. (a)). When $a$ is small enough, the tumor is controlled and the concentration of immune cells tends to the equilibrium value (fig. (a)). For larger $a$, the concentration of immune cells tends to a constant smaller than the expected equilibrium and the tumor mass blows up (fig. (b)-(d)). Mind that the time scale differs: in (b)-(d) the tumor mass mass blows rapidly, the larger $a$, the faster the blow up.

death rate induced by the effector immune cells precisely counterbalances the natural exponential growth of the tumor cell population.

Indeed, it is known that the growth-fragmentation operator admits an eigenpair $(\lambda, N)$ satisfying

$$\begin{cases} \partial_z(VN) - Q(N) + \lambda N = 0 \quad \text{for} \quad z \geq 0, \\ N(0) = 0, \qquad N(z) > 0 \quad \text{for} \quad z > 0, \qquad \int_0^{+\infty} N(z)\,\mathrm{d}z = 1, \qquad \lambda > 0. \end{cases} \tag{21}$$

We refer the reader to [35] for a detailed analysis of this eigenproblem. When the action of the immune system is neglected, namely $m(n, c) = 0$ and $b_1 = 0$ in (12a), the population of tumor cells grows exponentially fast and its size-distribution is governed by the eigenfunction: $n(t, z) \sim_{t \to \infty} e^{\lambda t} N(z)$, see [26–28]. Equilibrium occurs when the death rate due to the effector cells reaches the eigenvalue. Namely, the concentration $C$ of cytotoxic effector cells at

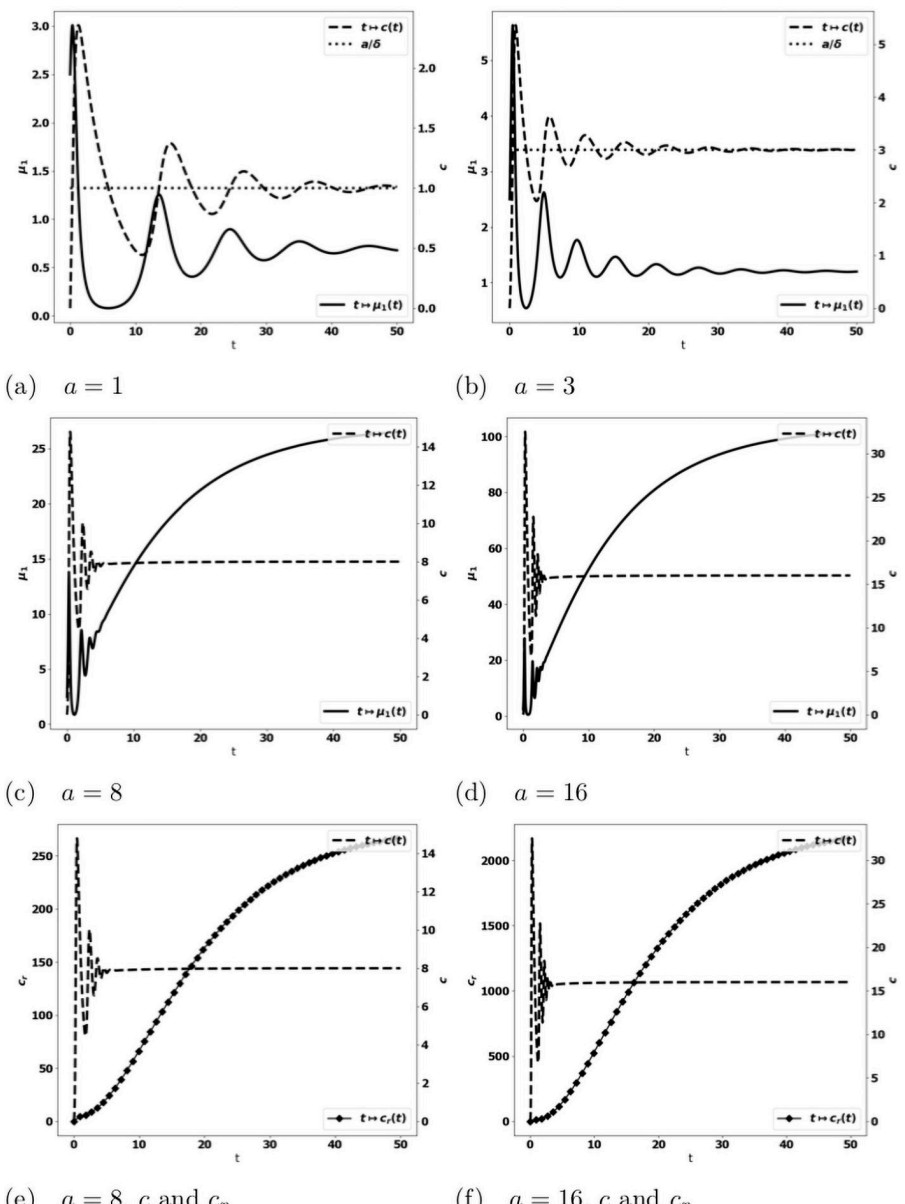

(a)   $a = 1$

(b)   $a = 3$

(c)   $a = 8$

(d)   $a = 16$

(e)   $a = 8$, $c$ and $c_r$

(f)   $a = 16$, $c$ and $c_r$

**Fig 4. Typical behavior of the solutions of the simplified ODE system (13) with $g(\mu_1) = \mu_1^2$. Data: $V = 0.616$, $\delta = 1.$,** $S = 1.5$, $k_r = 1.25$, $k_c = 0.1$, $m = 2$. Fig. (a)-(d): x-axis: time, y-axis: $\mu_1$, mass of the tumor (plain, left axis), and $c$, strength of the active immune cells (dashed, right axis), the expected equilibrium value is $a/\delta$ (dotted line in fig. (a) and (b)). Fig. (e)-(f): x-axis: time, y-axis: $c$ strength of the active immune cells (dashed, right axis), $c_r$, concentration of protumor cells (square, left axis). The equilibrium is restored with a strengthened recruitment of antitumor immune cells for large tumor masses. The behavior of the concentration of protumor cells follows the behavior of the tumor mass (compare (c)-(e) or (d)-(f)).

equilibrium should satisfy

$$\int_{\Omega} \delta(x) C(x)\, \mathrm{d}x = \lambda. \tag{22}$$

In turn, the definition of the concentration of activated immune cells by means of stationary convection-diffusion-reaction equations defines implicitly the total tumor mass at equilibrium. This intuition is made precise by the following statement which extends the results of [24] to the situation with protumor immune activities and inflammatory signals.

**Theorem 1** *Let $\Phi$ be the solution of*

$$\nabla_x \cdot (\mathcal{K} \nabla_x \Phi) = \sigma,$$

*endowed with the homogeneous Neumann boundary condition. There exists $\ell_\star > 0$ such that for any $0 < \ell < \ell_\star$, there exists a unique $\bar{\mu}_1(\ell) > 0$ and $(C_{\bar{\mu}_1(\ell)}, C_{r,\bar{\mu}_1(\ell)}, I_{\bar{\mu}_1(\ell)})$, solution of the stationary equations*

$$\gamma C + k_r I \theta C + k_c C C_r - f(\bar{\mu}_1) \nabla_x \cdot (C \chi \nabla_x \phi) - \nabla_x \cdot (D \nabla_x C) = g(\bar{\mu}_1) S, \qquad (23a)$$

$$\gamma_r C_r - f(\bar{\mu}_1) \nabla_x \cdot (C_r \chi \nabla_x \phi) - \nabla_x \cdot (D \nabla_x C_r) = I(S_r + k_r \theta C), \qquad (23b)$$

$$I = \frac{\psi(\bar{\mu}_1)}{\tau}, \qquad (23c)$$

$$C|_{\partial\Omega} = 0, \quad C_r|_{\partial\Omega} = 0, \qquad (23d)$$

*satisfying $\int_\Omega \delta C \, \mathrm{d}x = \ell$.*

**Proof.** We adapt the arguments from [24], taking into account protumor immune cells and cytokines. We start by introducing the mapping

$$\mathscr{F} : (\ell, \mu_1) \in [0, \infty) \times [0, \infty) \mapsto \int_\Omega \delta C_{\mu_1} \, \mathrm{d}x - \ell$$

where $C_{\mu_1}$ is the solution of (23a) associated to $\mu_1$. We are searching for the zeroes of $\mathscr{F}$. Clearly, when $\mu_1 = 0$, $C_0 = 0$, $I_0 = 0$, $C_{r,0} = 0$ satisfies (23a)–(23d), together with the constraint $\int \delta C_0 \, \mathrm{d}x = 0$, so that $\mathscr{F}(0, 0) = 0$. Next, we have $\partial_{\mu_1} \mathscr{F}(\ell, \mu_1) = \int_\Omega \delta C'_{\mu_1} \, \mathrm{d}x$, where $C'_{\mu_1}$ is defined by the system

$$(\gamma + k_r I \theta + k_c C_r) C' + (k_r I' \theta + k_c C'_r) C - \nabla_x \cdot (D \nabla_x C') - f(\mu_1) \nabla_x \cdot (C' \nabla_x \Phi)$$
$$= g'(\mu_1) S + f'(\mu_1) \nabla_x \cdot (C_{\mu_1} \nabla_x \Phi)$$

$$\gamma_r C'_r - \nabla_x \cdot (D \nabla_x C'_r) - f(\mu_1) \nabla_x \cdot (C'_r \nabla_x \Phi),$$
$$= I'(S_r + k_r \theta C) + I k_r \theta C' + f'(\mu_1) \nabla_x \cdot (C_{\mu_1} \nabla_x \Phi),$$

$$I' = \frac{\psi'(\mu_1)}{\tau}.$$

With $\mu_1 = 0$, the right hand side of the equation for $C'_r$ vanishes and we get $C'_r = 0$. In the right hand side of the equation for $C'$, $g'(0)S \neq 0$ is non negative and the maximum principle for elliptic equations tells us that $C'_0 > 0$. It follows that $\partial_{\mu_1} \mathscr{F}(0, 0) = \int_\Omega \delta C'_0 \, \mathrm{d}x > 0$. We can thus apply the implicit function theorem: there exists $\ell_\star > 0$ and a mapping $\bar{\mu}_1 : \ell \in [0, \ell_\star) \mapsto \bar{\mu}_1(\ell)$ such that $\mathscr{F}(\ell, \bar{\mu}_1(\ell)) = 0$ holds for any $\ell \in [0, \ell_\star)$. We have

$$\partial_\ell \mathscr{F}(\ell, \bar{\mu}_1(\ell)) + \bar{\mu}'_1(\ell) \partial_{\mu_1} \mathscr{F}(\ell, \bar{\mu}_1(\ell)) = -1 + \bar{\mu}'_1(\ell) \partial_{\mu_1} \mathscr{F}(\ell, \bar{\mu}_1(\ell)) = 0$$

with $\partial_{\mu_1} \mathscr{F}(0, 0) > 0$. Hence, $\ell \mapsto \bar{\mu}_1(\ell)$ is increasing on the neighborhood of $\ell = 0$, and it thus takes positive values.

Theorem 1 applies directly when the action of protumor immune cells is neglected on the cell-division equation (namely assuming $b_1 = 0$): the eigenstate $(\lambda, N)$ is defined a priori and the statement directly defines the equilibrium phase with the constraint (22). The statement involves a smallness assumption and it justifies the existence of equilibria with small tumor masses. In the case of the binary division model with a constant division rate $a$ and a constant growth rate $V$, the smallness assumption is equivalent to a smallness assumption on the division rate $a$. The numerical simulations indicate a quite robust property [24, 36] and the smallness assumption could be only technical.

The analysis of the full model accounting for a protumor action on the tumor growth rate is much more involved: $V(z)$ is multiplied by the factor $(1 + \beta(t))$ with

$$\beta(t) = \int b_1(y) c_r(t, y) \, dy.$$

As $t \to \infty$, we expect that $c_r(t, x)$ and $c(t, x)$ admits limits so that $\int \delta c(t, x) dx$ tends to some $\lambda > 0$ and $\beta(t)$ tends to an asymptotic value $\beta_\infty$ while the size-distribution of the tumor cells is described by an eigenpair $(\lambda, N)$. However, in general the eigenvalue $\lambda$ depends on the value of $\beta_\infty$, which induces a stronger coupling between the unknowns. The case where both $a$ and $V$ are constant is specific and allows us to strengthen the intuition. In this case, the leading eigenvalue $\lambda = a$ does not change when $V$ is replaced by $V(1 + \beta_\infty)$; only the profile is rescaled into

$$N_{\beta_\infty}(z) = N_1\left(\frac{z}{1 + \beta_\infty}\right),$$

where the profile $N_1$ is known. As $\beta_\infty$ increases, the asymptotic size-distribution of tumor cells contains larger cells, see [24, Fig. 3]. For general coefficients, the leading eigenstate of the growth-fragmentation operator is not explicitly known but it can be computed numerically using the power method designed in [36]: Fig 5 gives the value of the eigenvalue $\lambda$ and shows the profiles of the eigenfunctions when the growth rate $z \mapsto V(z)$ follows the Gompertz law (1) and the division rate is given by $z \mapsto a(z) = a \mathbf{1}_{z_0 \leq z < \infty}$ for some $z_0 > 0$. Changing $V$ now modifies both the profile and the eigenvalue.

For the same reasons, when the protumor cells modify the growth rate of the tumor cells, we cannot use as such the method designed in [36] to predict the equilibrium state. Nevertheless, the numerical simulations of the initial-boundary value problem highlight the following features:

- when the critical mass $m$ is positive, either the steady state is free of protumor immune cells ($c_r \equiv 0$) and the tumor is controlled by the antitumor immune cells or both the tumor mass $\mu_1$ and the concentration of protumor immune cells $c_r$ blow up. The former occurs for small division rates, the latter is observed with more aggressive tumors. Furthermore, when the tumor growth is controlled, we can check that the asymptotic concentration of antitumor immune cells ($x \mapsto C(x)$) satisfies (22) (see the numerical experiments detailed below)

- when the critical mass is equal to zero ($m = 0$), either we observe an equilibrium state containing residual protumor immune cells or the tumor mass $\mu_1$ and the concentration of protumor immune cells $c_r$ blow up.

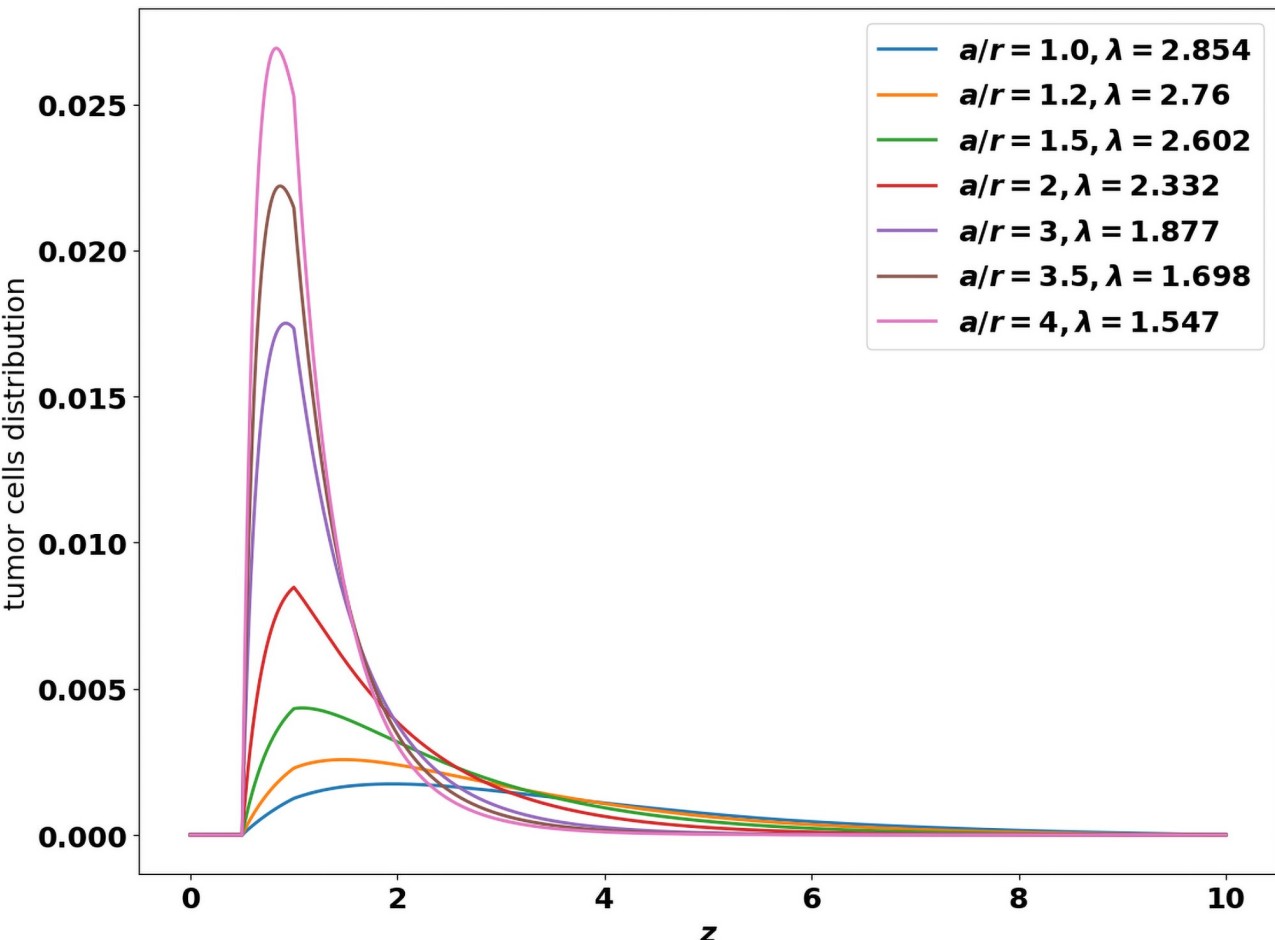

**Fig 5. Shapes of the leading eigen-function, solution of the growth-division Eq (21).** x-axis: $z$, size of the tumor cells, y-axis: number of tumor cells at the final time. The profiles are obtained for the binary cell division operator (3) with a constant division rate $a > 0$. and the Gompertz law (1) for the growth rate. The shape is driven by the ratio $\frac{a}{r}$ where $r = (1 + \beta_\infty)$ is the intrinsic growth rate of the tumor cells.

## Results

This section is devoted to numerical investigations of the PDE system (12a)–(12g). In the absence of protumor immune effects, the establishment of an equilibrium seems to be a quite robust feature of the model [24, 36]. This raises the issue to determine whether or not the protumor effects can break the equilibrium and lead to the escape phase and what are the key parameters to such a transition. We start by discussing this issue, dealing with quite complex growth-division coefficients, and equally with simpler constant coefficients, for which the details of the equilibrium are explicitely known, in order to fully validate the intuition on numerical grounds. The next challenge is to incorporate in the model the effects of treatments in order to understand how they can help in avoiding the escape phase and restore an equilibrium. To this end, we shall need to introduce additional equations describing exhausted immune cells the antitumor activity of which can be restored by the treatments. Another possibility is to reduce the protumor effects of the cytokine signals. We are going to discuss the effects of both strategy, bringing out the role of the administration dose and the starting time of the treatment. Finally, we show that combining the two approaches is highly beneficial, providing a better control of the tumor mass, with reduced dose and later administration.

**Table 2. Data for the simulations.**

| $A$ | $\xi^2$ | $A_\sigma$ | $\xi_\sigma^2$ | $a$ | $V$ | $r$ | $b$ | $\chi$ | $S$ | $\gamma$ |
|---|---|---|---|---|---|---|---|---|---|---|
| 1 | 0.02 | 0.002 | 0.05 | 0.8 | 0.616 | 0.616 | 10 | 0.864 | 5 | 0.18 |

| $A_{b_1}$ | $\xi_{b_1}$ | $A_\theta$ | $\xi_\theta$ | $k_r$ | $k_c$ | $\tau$ | $\eta$ | $m$ |
|---|---|---|---|---|---|---|---|---|
| $10^{-6}$ | 0.03 | 0.2 | 0.3 | 1 | 1 | 1 | 1 | 2 |

The mathematical model (12a)–(12g) accounts for the action of both anti- and protumor immune cells shaping tumor growth kinetics. Numerical experiments were used to challenge the model. We perform the numerical simulations considering the binary division operator (3) with a constant division rate $a > 0$. For details on the numerical methods, which are based on finite volume discretizations, we refer the reader to [24, 36]. According to the framework in [24], we assume that the tumor is located at the origin of the computational domain $\Omega$, which here is the unit ball of $\mathbb{R}^2$, and we use the following definitions

$$\delta(x) = \frac{A}{\xi\sqrt{2\pi}} \exp\left(-\frac{|x|^2}{2\xi^2}\right), \quad b_1(x) = \frac{A_{b_1}}{\xi_{b_1}\sqrt{2\pi}} \exp\left(-\frac{|x|^2}{2\xi_{b_1}^2}\right). \tag{24}$$

For defining the source term of the chemoattractant potential and the form function $\theta$ we also use the following Gaussian profiles:

$$\sigma(x) = \frac{A_\sigma}{\xi_\sigma\sqrt{2\pi}} \exp\left(-\frac{|x|^2}{2\xi_\sigma^2}\right), \quad \theta(x) = \frac{A_\theta}{\xi_\theta\sqrt{2\pi}} \exp\left(-\frac{|x|^2}{2\xi_\theta^2}\right) \tag{25}$$

In what follows, we use the Michaelis-Menten law (7) and we simply set $g(\mu_1) = \mu_1$. For the simulations, we shall use the following data, otherwise explicitly stated: the initial data are $(c_0(x), c_{r,0}(x)) = (0, 0)$ and $n_0(z) = \mathbf{1}_{0.125 \leq z \leq 5}$. The parameters are given in Table 2. We use dimensionless equations, without addressing precisely the issue of parameter calibration (see [24, 36] for further details on this issue). For the numerical experiments, we consider the source of antitumor cells, that contains T cells recruited in specific sites like the lymph nodes, as well as NK, N1, M1 taken from the circulation, as space-homogeneous. In contrast, we work with an heterogeneous source $S_r$ of protumor cells, distant from the tumor site (see Fig 6; in practice, we use 1/5 of the source depicted in this figure). This assumption corresponds to a privileged recruitment of protumor cells in specific sites, like the bone marrow.

## Emergent qualitative features: Promotumor cells are determinant for the shift to the escape phase

In order to assess the qualitative behavior of the system (12a)–(12g), we perform a set of simulations with assumptions that incorporate several biological effects, namely

- $z \mapsto V(z)$ obeys the Gompertz law (1);

- the division rate $z \mapsto a(z) = a\mathbf{1}_{z_0 \leq z < \infty}$ for some $z_0 > 0$ (for the numerical tests we set $z_0 = 1$) vanishes for the smallest cells;

- the presence of the protumor immune cells promotes tumor growth with $b_1 \neq 0$.

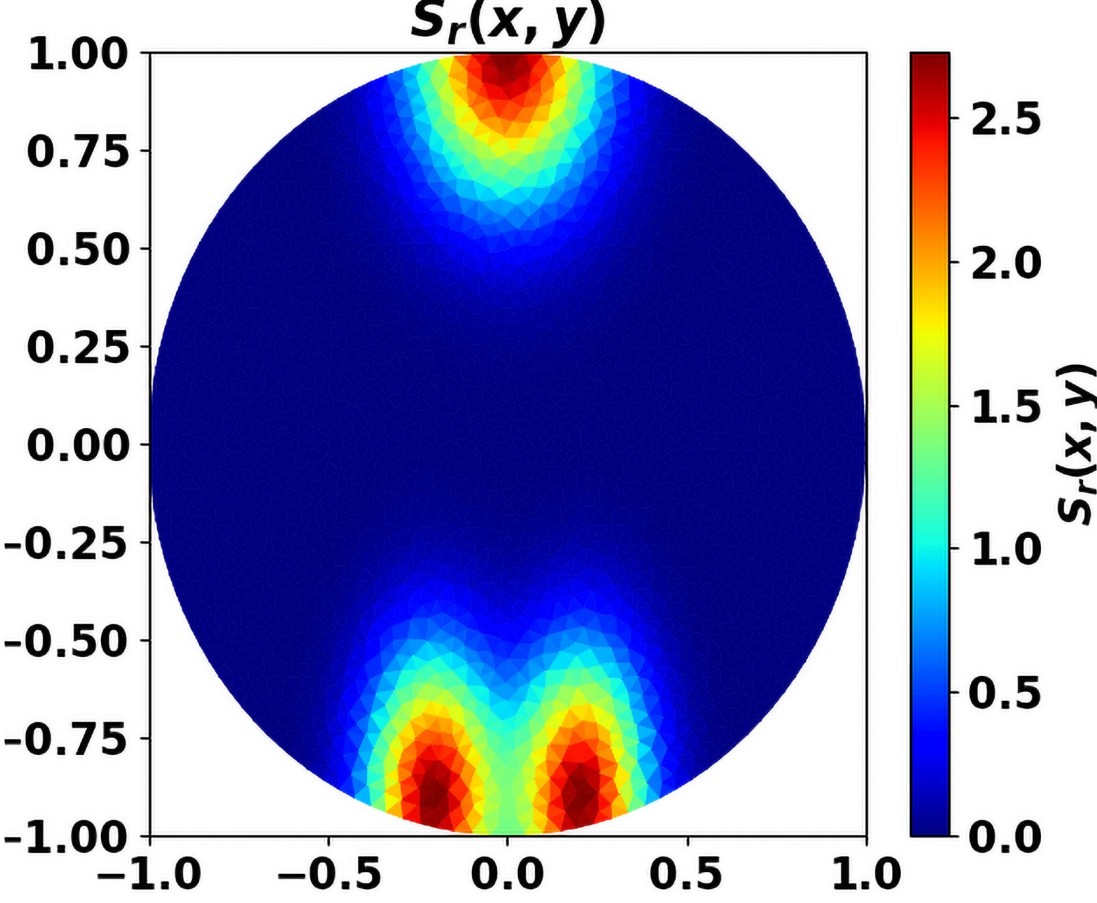

**Fig 6. Source $S_r$ of protumor cells.** x-y axis correspond to the space coordinates.

We generically observe two behaviors: either an equilibrium state establishes, with a residual tumor and free of protumor immune cells, or the immune system fails in controlling the tumor, with a significant concentration of protumor immune cells at the center of the domain (like in Fig 7 which also clearly illustrates how the antitumor resources is stemmed in the vicinity of the tumor) and the tumor mass blows up. This rough conclusion should be nuanced: the threshold $m$ certainly plays a critical role. With $m = 0$, the worst situation since protumor immune cells are immediately activated, we can find equilibria with the three types of cells (tumor cells, antitumor and protumor immune cells). Such equilibria occur with quite small values of the cell division rate $a$; increasing $a$ leads to an escape phase. It is likely that similar phenomena occur with positive threshold $m$ and very small $a$'s. When a steady regime establishes, we check on numerical ground, by evaluating the eigenpair of the growth-division equation, that the asymptotic antitumor cell concentration is consistent with the constraint (22).

We make the parameters vary in order to discuss the influence of their value on the behavior of the system. We only modify one quantity at a time, the others being kept as in Table 2.

- *Tumor aggressiveness*. As indicated in [24], by increasing the division rate $a$, we make the tumor more aggressive. The results with the PDE system are consistent with this intuition and the observations made above on the simplified ODE system: for small division rates, the mass of the tumor is rapidly damped, while the tumor escapes the control of the immune

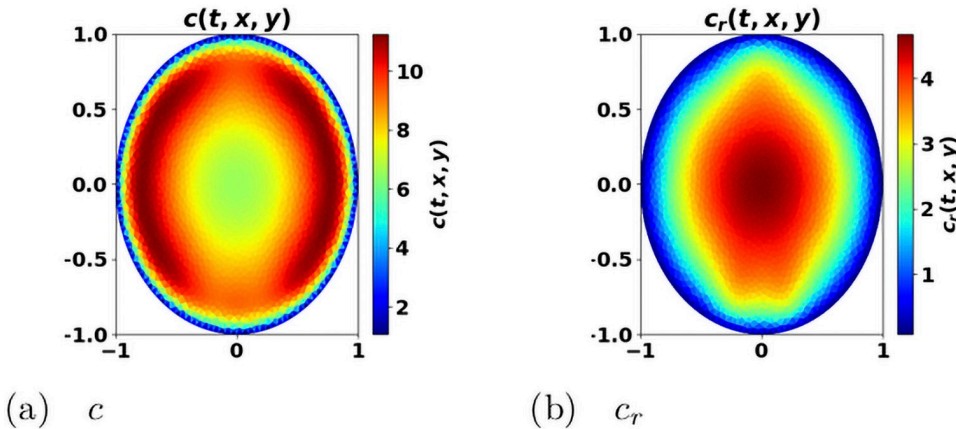

**Fig 7. Space distribution of immune cells.** Antitumor cells $c$ (left) and protumor cells $c_r$ (right). x-y axis: space coordinates. The concentration of protumor cells is higher in the vicinity of the tumor, where the concentration of antitumor cells has been depleted. The picture has been obtained from the simulation of the system (12a)–(12g), at final time $t = 2.23$, with $a = 4$.

system as $a$ increases, see Fig 8. When control occurs, protumor immune cells can be activated in the transient states, but insufficiently to counterbalance the effector immune response. Therefore, the concentration of protumor immune cells decreases to zero, while

$$\bar{\mu}_c(t) = \int_\Omega \delta(x) c(t, x) \, \mathrm{d}x \tag{26}$$

tends to the leading eigenvalue $\lambda$ of the growth-division equation, see Figs 8 and 14a. When the tumor is more aggressive, it recruits more protumor immune cells: in turn, the action of these cells restrains the concentration of antitumor immune cells which remains below the expected equilibrium value, eventually favoring the tumor escape. The specific value of $a$ for the bifurcation from a controlled tumor growth to the escape state depends on the critical mass $m$: the smaller the critical mass $m$, the smaller the critical division rate $a$. Figs 9 and 10 illustrate the space organization of the immune response in a situation where the control of the tumor is lost (the data are the same as in Fig 8(c)). Antitumor cells are activated and recruited from a homogeneous bath while protumor cells come from specific spots, as it appears clearly in the earliest pictures. Both are directed towards the tumor, at the center of the domain, by the chemotactic potential. In the situation depicted in these figures, the protumor action becomes strong in the vicinity of the tumor where it inhibits the antitumor response.

- *Efficiency of the immune response.* The immune response is enhanced by increasing $A$, that measures the strength of the immune cells against the tumor cells, see (24), or the source $S$ of effector antitumor cells. For small values of these parameters, the tumor escapes the control of the immune response, see Figs 11 and 12. While the tumor mass $\mu_1$ keeps growing, the immune strength $\bar{\mu}_c$ remains limited and cannot balance the growth rate of the tumor. For large $S$ or $A$, the equilibrium establishes, with $\bar{\mu}_c(t)$ tending asymptotically to counterbalance the tumor growth rate $\lambda$.

**Further numerical validation.** The case where the growth rate $V$ and the division rate $a$ are constant is less relevant biologically, but it can be used to check the properties of the model

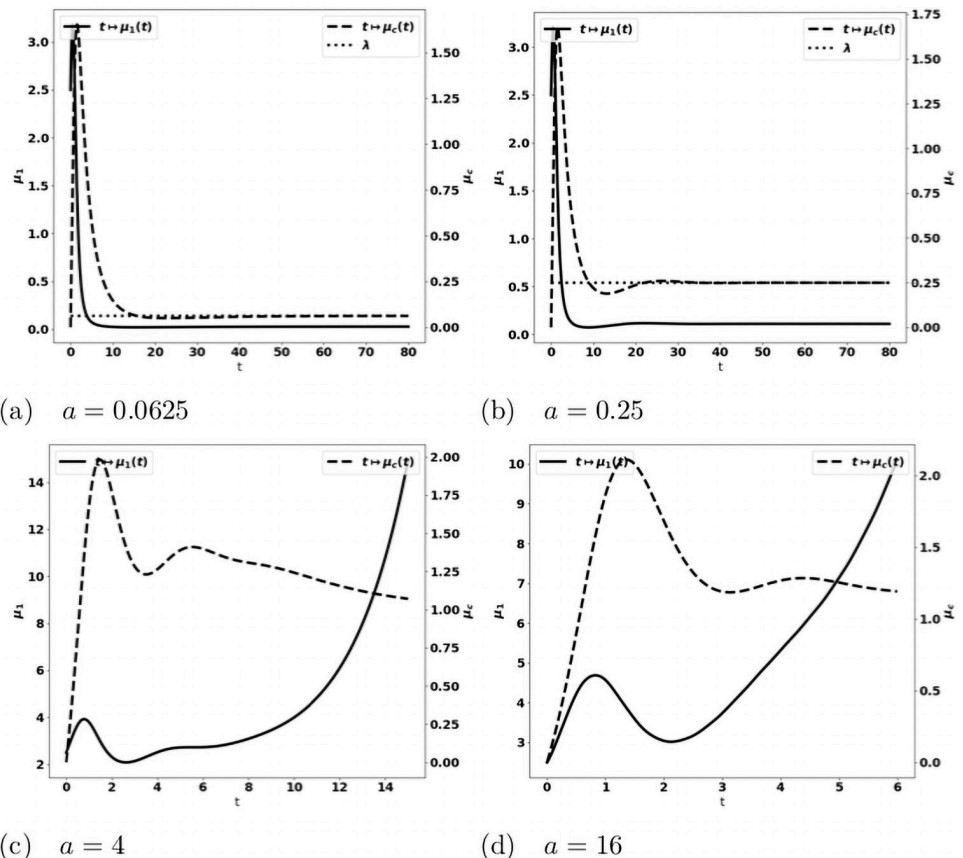

**Fig 8. Simulation of the PDE model (12a)–(12g) for several values of the division rate *a*.** Evolution of the tumor mass $\mu_1$ (plain, left axis), and of the immune strength $\bar{\mu}_c$ (defined in (26), dashed, right axis); expected equilibrium value λ (dotted line in fig. (a) and (b)). When *a* is small enough an equilibrium is reached with $\bar{\mu}_c$ tending to the eigenvalue λ of the cell-division equation, and a residual tumor mass (fig. (a) and (b)). For larger *a*'s the tumor escapes the control of the immune system and its mass bows up (fig. (c) and (d)): mind the different time scales, since the blows up is faster as *a* increases.

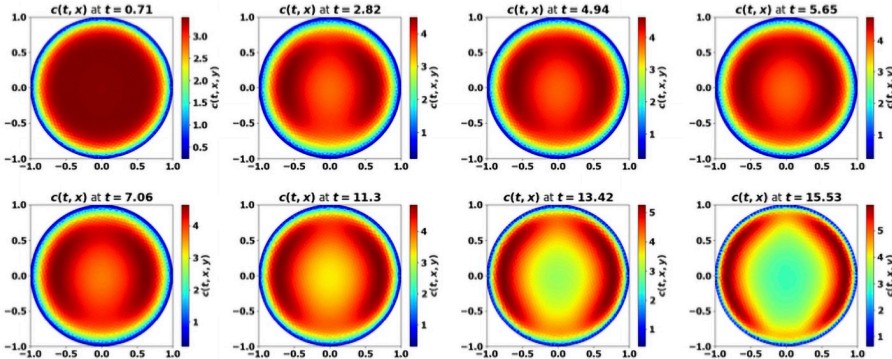

**Fig 9. Simulation of the PDE model (12a)–(12g): Evolution of the size repartition of the antitumor immune cells.** The test corresponds to Fig 8(c), with *a* = 4. x-y axis: space coordinates. The antitumor cells are recruited from a homogeneous bath of non-activated cells, and then are directed towards the tumor by chemotactism. However, the protumor activities lower their concentration in the vicinity of the tumor (mind that the scale of the color map changes with time).

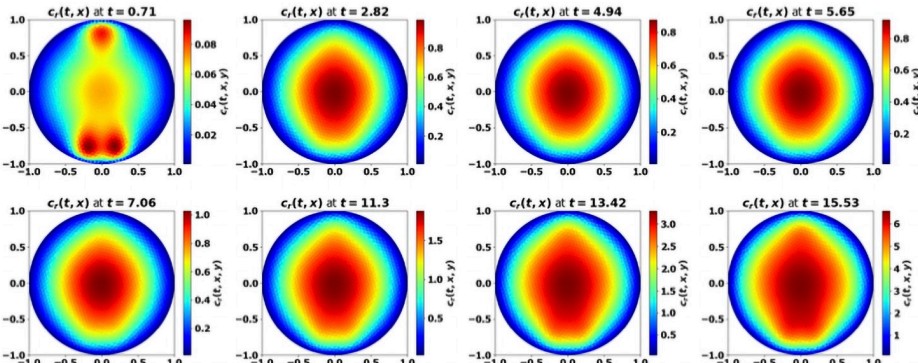

**Fig 10. Simulation of the PDE model (12a)–(12g): Evolution of the size repartition of the protumor immune cells.**
The test corresponds to Fig 8(c), with $a = 4$. The protumor cells are activated in 3 specific sites, and then are directed towards the tumor by chemotactism. Eventually there is a high concentration of protumor cells in the vicinity of the tumor (mind that the scale of the color map changes with time).

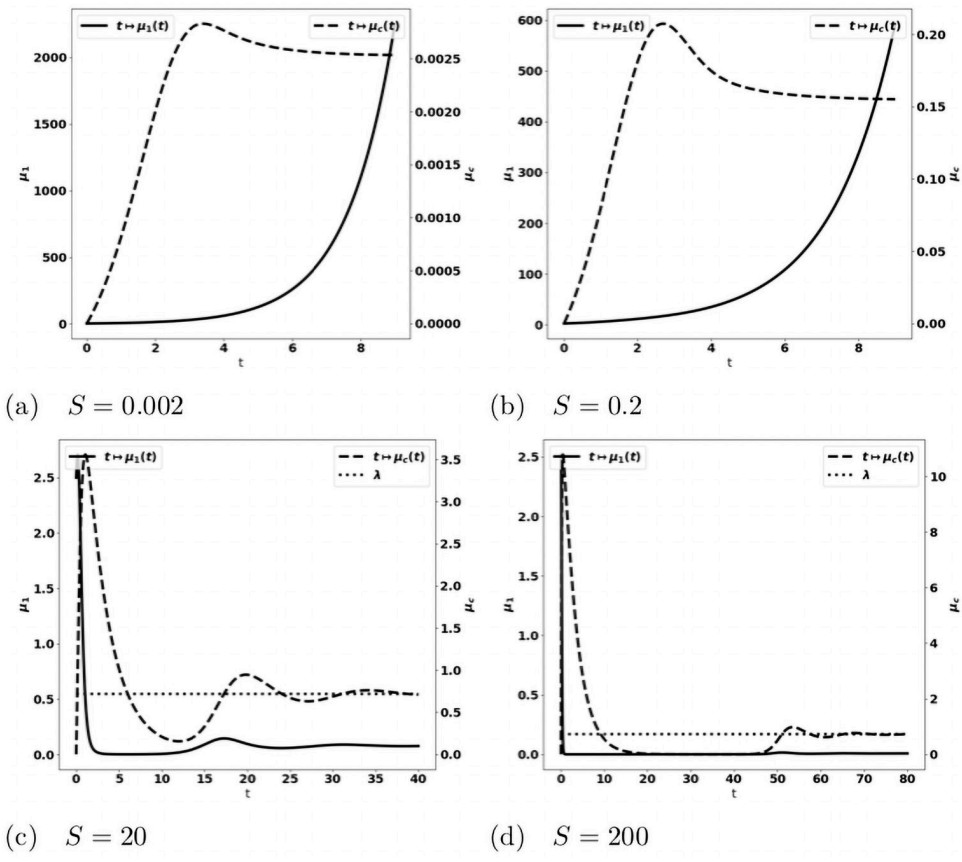

**Fig 11. Simulation of the PDE model (12a)–(12g) for several values of the source of effector immune cells S.**
Evolution of the tumor mass $\mu_1$ (plain, left axis), and of the immune strength $\bar{\mu}_c$ (defined in (26), dashed, right axis). When $S$ is large enough an equilibrium is reached with $\bar{\mu}_c$ tending to the eigenvalue $\lambda$ of the cell-division equation (dotted line in fig. (c) and (d)), and a residual tumor mass (fig. (c) and (d)). For smaller $S$'s the tumor escapes the control of the immune system and its mass blows up (fig. (a) and (b)).

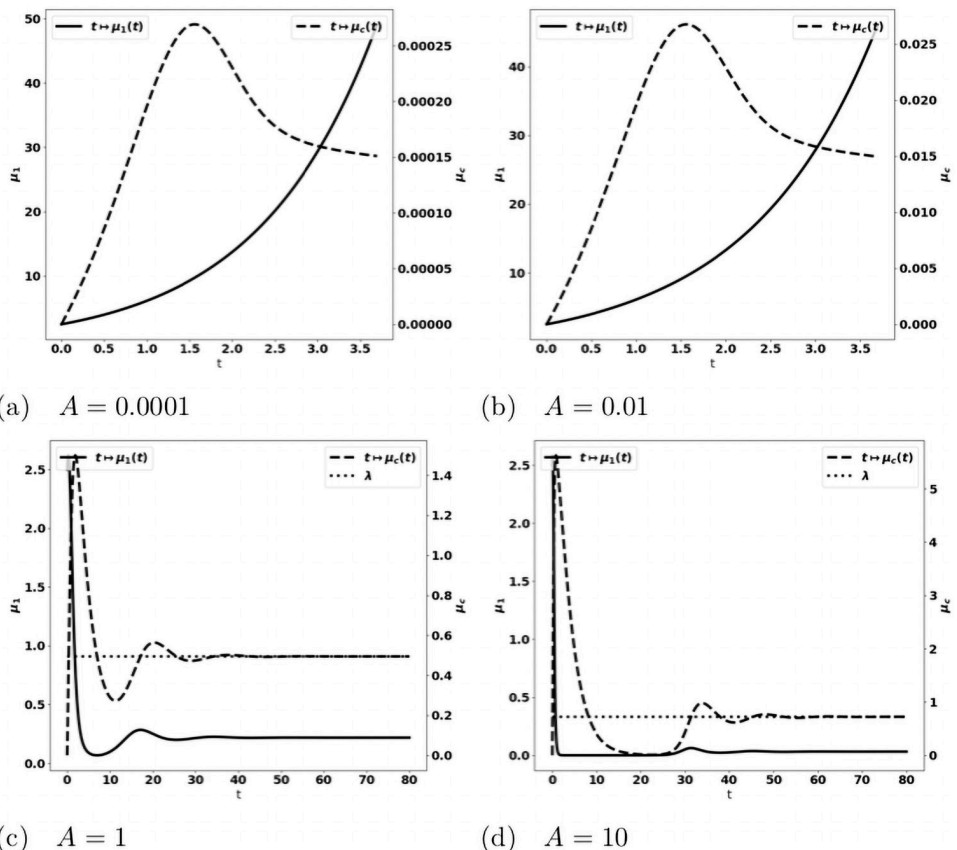

**Fig 12. Simulation of the PDE model (12a)–(12g) for several values of the immune strength $A$.** Evolution of the tumor mass $\mu_1$ (plain, left axis), and of the immune strength $\bar{\mu}_c$ (defined in (26), dashed, right axis); expected equilibrium value $\lambda$ (dotted line in fig. (c) and (d)). When $A$ is large enough an equilibrium is reached with $\bar{\mu}_c$ tending to the eigenvalue $\lambda$ of the cell-division equation, and a residual tumor mass (fig. (c) and (d)). For smaller $A$'s the tumor escapes the control of the immune system and its mass bows up (fig. (a) and (b)).

and of the numerical procedures since the eigenpair $(\lambda, N)$ is explicitly known in this case, see [37, 38] and [28, Lemma 4.1]: in fact we have $\lambda = a$. Still with the purpose of assessing the model and the numerical method on simple basis, it is relevant to perform simulations by assuming also that the protumor immune cells do not enhance the growth rate of the tumor cells ($b_1 = 0$). It means that the protumor effect is limited to the suppression of antitumor capacities. This case is biologically questionable, but, as explained above, we have an intuition a priori on the details of the possible equilibrium state, and we can directly check whether or not the large time behavior is described by this expected equilibrium.

Beyond the validation, it is remarkable that the qualitative conclusions do not substantially change when comparing the results to the more complete situation dealt before: compare Figs 8 and 11–15, respectively. The evolution of the concentration of protumor immune cells in Fig 16 is equally quite generic: we clearly see the difference between an equilibrium case where this concentration vanishes as time grows, and an escape case where it blows up. These observations show the robustness of the model in describing the equilibrium vs escape phenomena and this is very reassuring for further investigations with clinical data, as in [36]. In this direction, identifying the parameters of the equations is a critical issue. It can be interesting, based

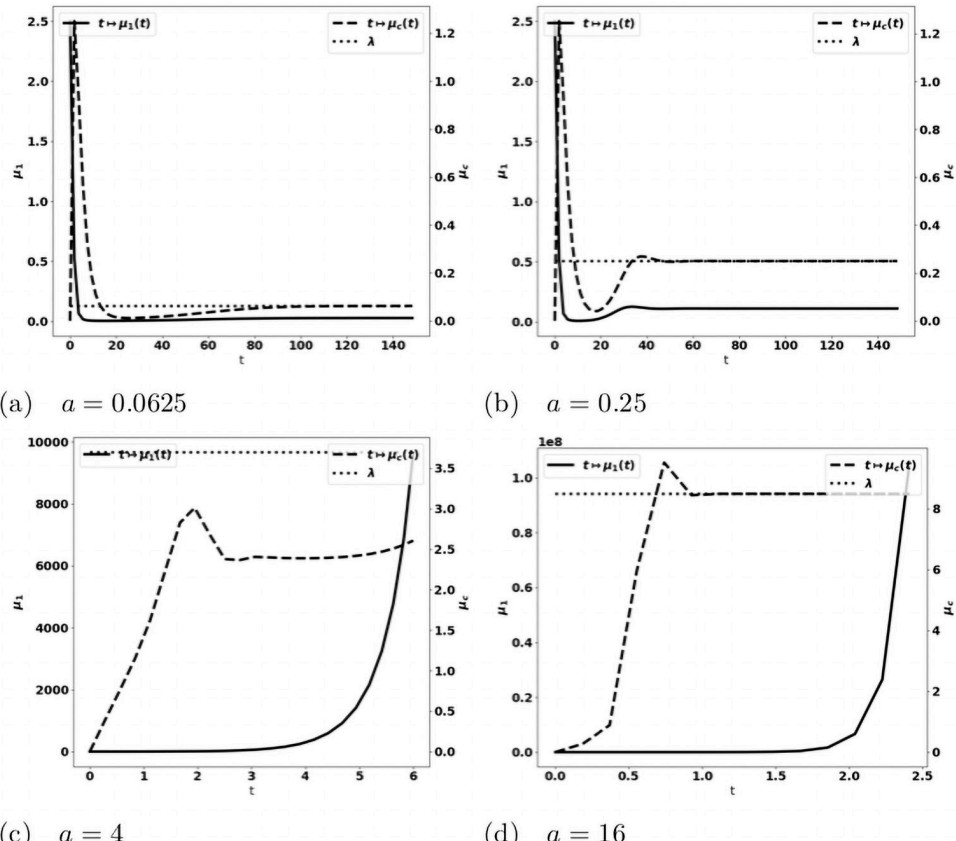

**Fig 13. Validation of the PDE model ($b_1 = 0$, $V$ and $a$ constant).** Evolution of the tumor mass $\mu_1$ (plain, left axis), and of the immune strength $\bar{\mu}_c$ (defined in (26), dashed, right axis) for several values of the division rate $a$; expected equilibrium value $\lambda$ (dotted line). When $a$ is small enough an equilibrium is reached with $\bar{\mu}_c$ tending to the eigenvalue $\lambda$ of the cell-division equation, and a residual tumor mass (fig. (a) and (b)). For larger $a$'s the tumor escapes the control of the immune system and its mass bows up (fig. (c) and (d)): mind the different time scales, since the blows up is faster as $a$ increases.

on the present observations, to neglect some phenomena which can only marginally affect the dynamics, while potentially introducing a set of unknown parameters.

## Effect of immunotherapy strategies

Now that we have validated a robust mathematical model of tumor growth which takes into account the contribution of anti and protumor immune cells, we use it to compare the effects of two immunotherapy treatments targeting these immune cells with opposed functions. To this end, we bear in mind that a proportion of the effector cells are just inhibited by immuno-suppressive mechanisms; in other words they are not destroyed: they become hyporesponsive. However, they can be re-activated by specific treatments. The restoration of the antitumoral activity of effector T cells can be obtained by using Immune Checkpoint Inhibitors, like anti-PD-1 or anti-CTLA4 antibodies [14, 15]. The infusion of CAR-T and CAR-NK cells can also mimic such rescue [39]. A second strategy is to reduce the recruitment of protumor immune cells by blocking infiltration of MDSCs (anti-CXCR2, cMet) [40, 41] and Tregs (anti-CD25) [13]. We discuss the effect of these approaches, illustrated in Fig 17, independently and we also consider the combination of the two treatments compared to the mono-therapies.

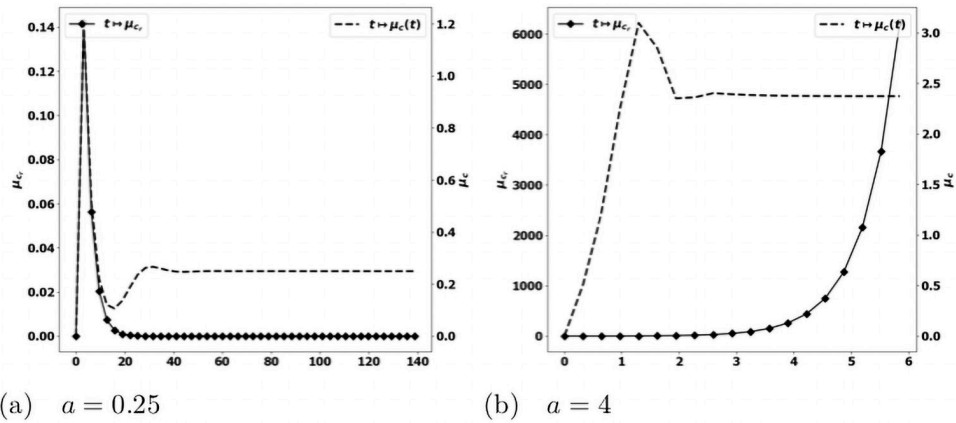

(a)  $a = 0.25$          (b)  $a = 4$

**Fig 14. Validation of the PDE model ($b_1 = 0$, $V$ and $a$ constant).** Evolution of the tumor mass $\mu_1$ (plain, left axis), and of the immune strength $\bar{\mu}_c$ (dashed, right axis) for several values of the source of immune cells $S$; expected equilibrium value $\lambda$ (dotted line in fig. (c) and (d)). When $S$ is large enough an equilibrium is reached with $\bar{\mu}_c$ tending to the eigenvalue $\lambda$ of the cell-division equation, and a residual tumor mass (fig. (c) and (d)). For smaller $S$'s the tumor escapes the control of the immune system and its mass bows up (fig. (a) and (b)).

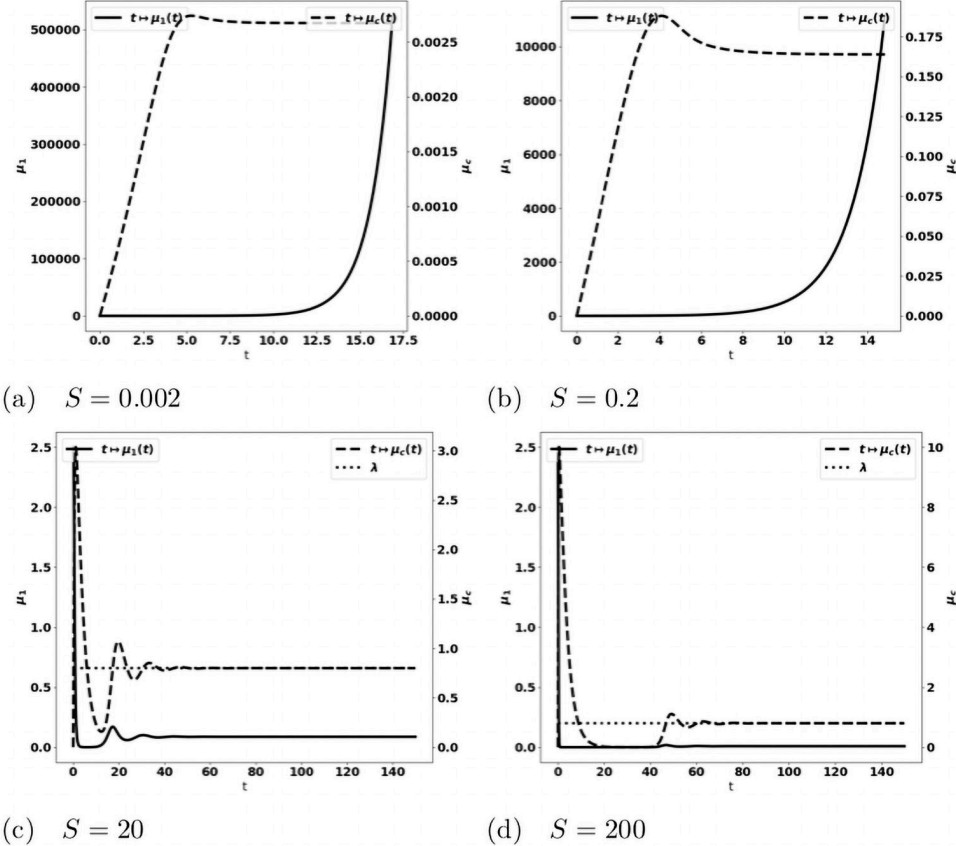

(a)  $S = 0.002$          (b)  $S = 0.2$

(c)  $S = 20$          (d)  $S = 200$

**Fig 15. Validation of the PDE model ($b_1 = 0$, $V$ and $a$ constant).** Evolution of the tumor mass $\mu_1$ (plain, left axis), and of the immune strength $\bar{\mu}_c$ (dashed, right axis) for several values of the source of immune strength $A$ expected equilibrium value $\lambda$ (dotted line in fig. (c) and (d)). When $A$ is large enough an equilibrium is reached with $\bar{\mu}_c$ tending to the eigenvalue $\lambda$ of the cell-division equation, and a residual tumor mass (fig. (c) and (d)). For smaller $A$'s the tumor escapes the control of the immune system and its mass bows up (fig. (a) and (b)).

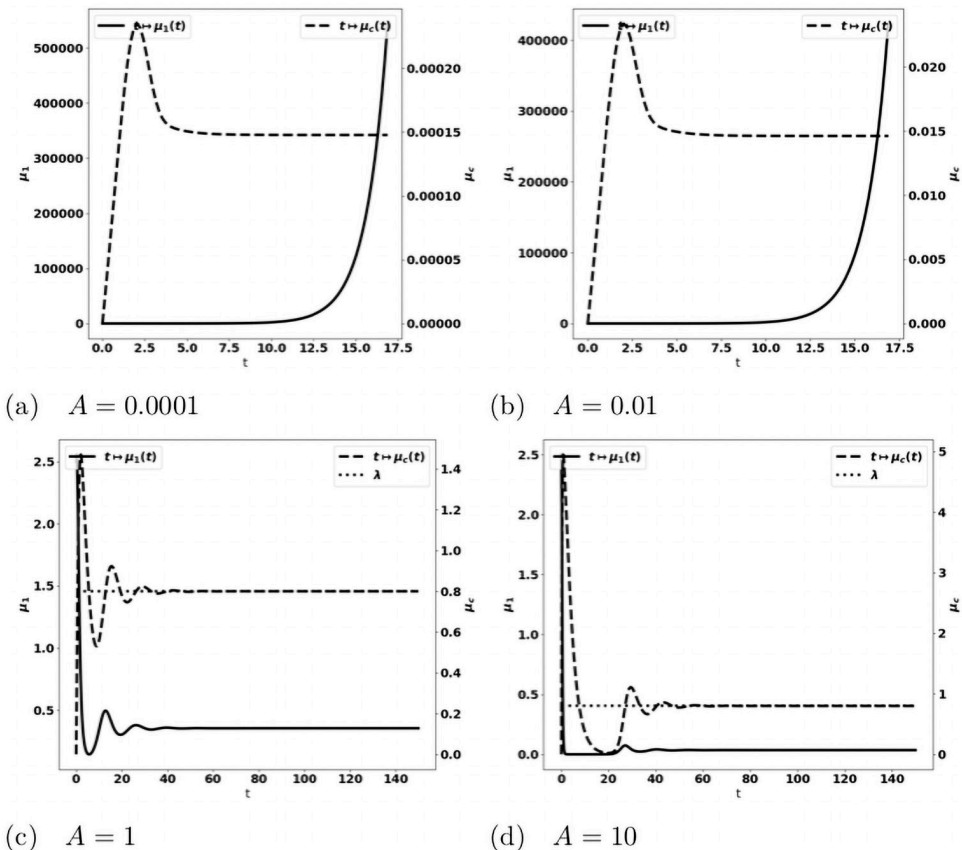

(a) $A = 0.0001$ (b) $A = 0.01$

(c) $A = 1$ (d) $A = 10$

**Fig 16. Validation of the PDE model ($b_1 = 0$, $V$ and $a$ constant).** Evolution of the averaged protumor cells concentration $\mu_{c_r} = \int_\Omega c_r \, dx$ (square, left axis), and of the immune strength $\bar\mu_c$ (dashed, right axis) for several values of the division rate $a$. For small $a$ (fig. (a)), the concentration of antitumor immune cells reaches the equilibrium, and the concentration of protumor cells is damped; for large $a$ (fig. (b)), the equilibrium does not establish and the concentration of protumor immune cells keeps growing.

**Therapy based on the reactivation of exhausted antitumor immune cells.** To consider the action of treatments boosting the immune response against the tumor, we introduce the concentration $c_a$ of exhausted cells. In order to describe the restoration mechanism, we add the following equation to the model:

$$\partial_t c_a + \nabla_x \cdot (c_a \chi \nabla_x \phi - D \nabla_x c_a) = \alpha k_c c c_r - \gamma_a c_a, \tag{27}$$

where the parameter $0 < \alpha < 1$ describes the proportion of effector T cells that become hyporesponsive under the action of the protumor cells (see (28a)). Note that the death rate $\gamma_a$ could be significantly larger than the original death rate $\gamma$: it is believed that exhausted T cells have a shorter life time. Next, the effect of treatments able to restore the antitumor activity of the exhausted immune cells is described by a time-dependent function $t \mapsto \mathcal{T}_1(t)$. It is assumed to be proportional to the drug concentration in the TME. Consequently, the dynamic is governed

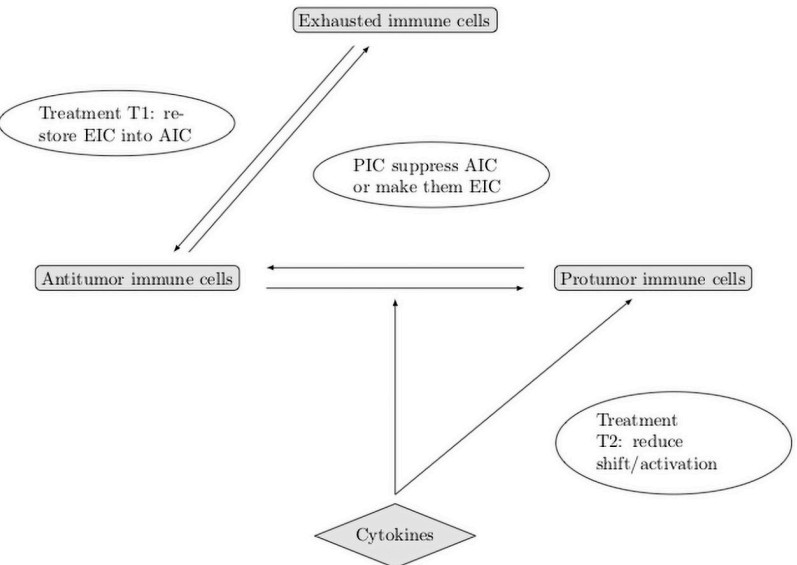

**Fig 17. Schematic overview of the action of treatments on the immune response.** AIC: antitumor cells, PIC: protumor cells, EIC: exhausted immune cells.

by the following system, which extends (12a)–(12g)

$$\partial_t c + \nabla_x \cdot (c\chi\nabla_x\phi - D\nabla_x c) = g(\mu_1)S + \mathcal{T}_1 c_a - \gamma c - k_r I\theta c - k_c cc_r, \tag{28a}$$

$$\partial_t c_r + \nabla_x \cdot (c_r\chi\nabla_x\phi - D\nabla_x c_r) = I(S_r + k_r\theta c) - \gamma_r c_r \tag{28b}$$

$$\partial_t c_a + \nabla_x \cdot (c_a\chi\nabla_x\phi - D\nabla_x c_a) = \alpha k_c cc_r - \gamma_a c_a - \mathcal{T}_1 c_a, \tag{28c}$$

$$\partial_t I = \psi(\mu_1) - \tau I \tag{28d}$$

$$-\nabla_x \cdot (\mathcal{K}\nabla_x\phi) = f(\mu_1)\sigma, \tag{28e}$$

$$c|_{\partial\Omega} = 0, \ \ c_r|_{\partial\Omega} = 0, \ \ \nabla_x\phi \cdot v(\cdot)|_{\partial\Omega} = 0, \tag{28f}$$

$$c(t=0, x) = c_0(x), \ \ c(t=0, x) = c_r^0(x), \ \ I(t=0) = I_0. \tag{28g}$$

The kinetic of the drug effect is described by the following equation

$$\partial_t \mathcal{T}_1 = \kappa(t) - d_{\mathcal{T}_1}\mathcal{T}_1, \tag{29}$$

where $t \mapsto \kappa(t)$ describes the drug administration protocol and $d_{\mathcal{T}_1}$ is the degradation rate of this drug. For the numerical tests, we set

$$\kappa(t) = \begin{cases} 0, & \forall 0 \leq t \leq t_0 \\ \sum_{k \geq 0} \kappa_k(t - kT_2), & \forall t \geq t_0 \end{cases} \tag{30}$$

where,

$$\kappa_k(t) = \begin{cases} q, & t_0 + kT_2 < t \leq t_0 + kT_2 + T_1 \\ 0, & t_0 + kT_2 + T_1 \leq t < t_0 + (k+1)T_2. \end{cases} \qquad (31)$$

The model depends on

- the time $t_0$ when the treatment starts,

- the duration $T_2$ between two drug administrations,

- the duration $T_1$ of the drug administration,

- the administered drug concentration $q$.

For the numerical tests, we place ourselves in the same configuration as in Fig 8(c) where the tumor escapes the immune control due to the effects of protumor immune cells. We fix $\alpha$, the proportion of effector T cells that become exhausted to 0.5 and we keep the other parameters as in Table 2. We set

$$T_1 = 1, \qquad T_2 = 7, \qquad d_{T_1} = 0.05$$

and we make the starting time $t_0$ and the dose $q$ vary. We indeed observe that these parameters have a critical role on the treatment efficacy.

When the treatment is given early (for instance, when $0 \leq t_0 \leq 5$), the control of the tumor can be obtained with relatively low drug doses (see Fig 18), in comparison to the cases where the treatment is administered later (Fig 19). At these early administration of the treatment, the tumor growth is controlled with a residue of dormant tumor cells and activated effector immune cells. Reducing the treatment dose reduces the drug efficacy with smaller tumor masses reached over longer period of time. For very small doses, the escape can occur.

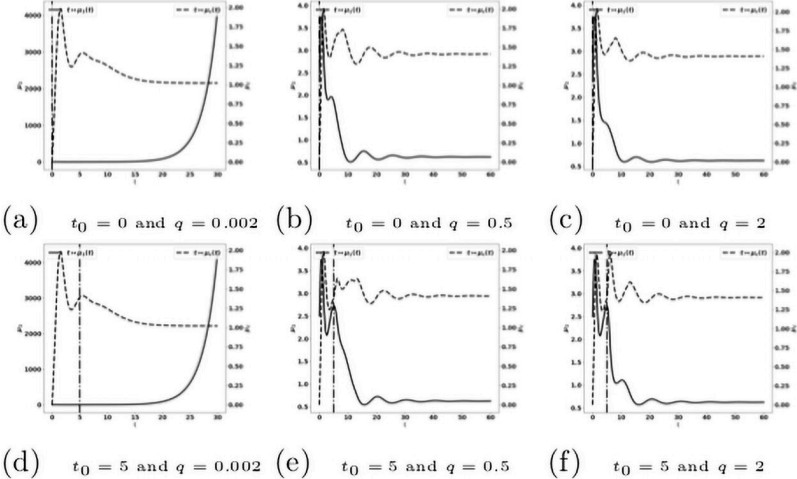

**Fig 18. Reactivation of exhausted antitumor cells: Early administration of the treatment.** Simulation performed on the system (28a)–(28g), with (29)–(31). Evolution of the tumor mass $\mu_1$ (plain, left axis), and of the immune strength $\bar{\mu}_c$ (dashed, right axis) for several values of the treatment dose $q$. The dash-dotted line represents the time at which the treatment starts.

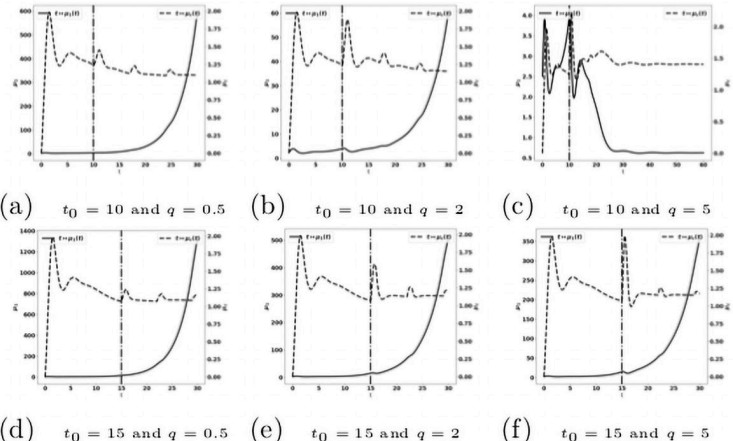

**Fig 19. Reactivation of exhausted antitumor cells: Late administration of the treatment.** Simulation performed on the system (28a)–(28g), with (29)–(31). Evolution of the tumor mass $\mu_1$ (plain, left axis), and of $\bar{\mu}_c$ (dashed, right axis) for several values of the treatment dose $q$. The dash-dotted line represents the time at which the treatment starts.

When the treatment is given later (for instance, when $10 \leq t_0 \leq 15$, see Fig 19): the tumor growth is slowed down by the treatment, but the tumor continues to grow exponentially fast. Increasing the drug dose increases the treatment efficacy. However, this observation raises the issue of the toxicity of the administered drug.

These observations are in line with experimental data using immune checkpoint inhibitors or CART/NK cells. Indeed, syngeneic CMS5 fibrosarcomas allowed to grow for 3 days in vivo were easily eradicated by adoptive transferred tumor-specific T cells while a 100-fold larger number of transferred tumor specific T-cell was mandatory to eradicate tumors that have been grown for an additional 48 hours. The same tumors that have been grown for 7 days before transferring adoptive tumor-specific T cells were not eradicated [42].

**Therapy based on reducing cytokine signals recruiting protumor immune cells.** Treatments based on blocking cytokine signals can help reducing the recruitment of protumor immune cells. A possible strategy uses cytokine traps [43, 44], by means of molecules that inhibit signal transduction from T cell cytokine receptors. Therefore, the treatment acts by down-regulating the effect of the tumor induced cytokines. We denote by $\mathcal{T}_2$, the effect of treatments which are able to block those cytokines. It obeys a kinetic similar to (29)

$$\partial_t \mathcal{T}_2 = \kappa^{(2)}(t) - d_{\mathcal{T}_2} \mathcal{T}_2, \tag{32}$$

where

$$\kappa^{(2)}(t) = \begin{cases} 0, & \forall 0 \leq t \leq t_0 \\ \sum_{k \geq 0} \kappa_k^{(2)}(t - kT_2), & \forall t \geq t_0 \end{cases} \tag{33}$$

and

$$\kappa_k^{(2)}(t) = \begin{cases} q_2, & t_0 + kT_2 < t \leq t_0 + kT_2 + T_1 \\ 0, & t_0 + kT_2 + T_1 \leq t < t_0 + (k+1)T_2. \end{cases} \tag{34}$$

The effect on the cytokines is described by modifying in (12b), (12c) the terms related to the cytokine-dependent recruitment of protumor immune cells. Therefore, the equations on the immune response become

$$\partial_t c + \nabla_x \cdot (c\chi\nabla_x\phi - D\nabla_x c) = g(\mu_1)S - \gamma c - k_r I[1 - \mathcal{T}_2]_+ \theta c - k_c c c_r, \tag{35a}$$

$$\partial_t c_r + \nabla_x \cdot (c_r\chi\nabla_x\phi - D\nabla_x c_r) = I[1 - \mathcal{T}_2]_+ (S_r + k_r\theta c) - \gamma_r c_r \tag{35b}$$

$$\partial_t c_a + \nabla_x \cdot (c_a\chi\nabla_x\phi - D\nabla_x c_a) = \alpha k_c c c_r - \gamma_a c_a, \tag{35c}$$

$$\partial_t I = \psi(\mu_1) - \tau I \tag{35d}$$

$$-\nabla_x \cdot (\mathcal{K}\nabla_x\phi) = f(\mu_1)\sigma, \tag{35e}$$

$$c|_{\partial\Omega} = 0, \;\; c_r|_{\partial\Omega} = 0, \;\; \nabla_x\phi \cdot v(\cdot)|_{\partial\Omega} = 0, \tag{35f}$$

$$c(t = 0, x) = c_0(x), \;\; c(t = 0, x) = c_r^0(x), \;\; I(t = 0) = I_0. \tag{35g}$$

For the numerical tests, we set

$$T_1 = 1, \qquad T_2 = 7, \qquad d_{T_2} = 0.0105.$$

For the cytokine-blockade based treatment we observe a similar behavior as with the treatment based on the reactivation of the exhausted immune cells. The efficacy of the treatment is particularly sensitive to the starting time $t_0$, see Fig 20.

**Combination of the two immunotherapy strategies.** When we combine the two treatments described above, acting on both the reactivation of antitumor immune cells and the blockade of the recruitment of protumor immune cells, we observe that this combination is

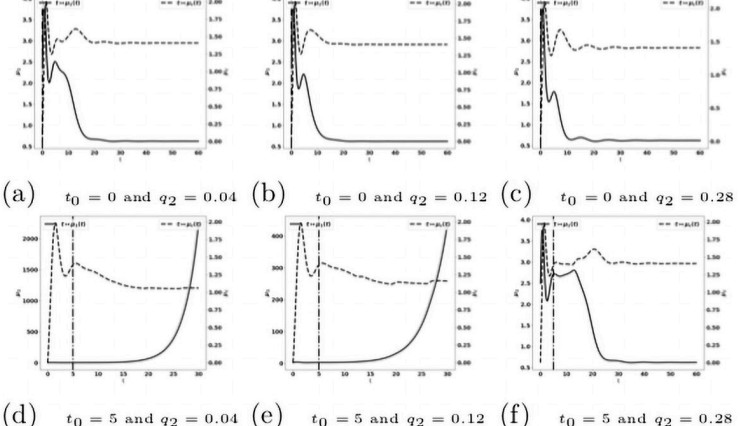

**Fig 20. Reduction of the protumor recruitment: Early administration of the treatment.** Simulation performed on the system (35a)–(35g), with (32)–(34). Evolution of the tumor mass $\mu_1$ (plain, left axis), and of the immune strength $\bar{\mu}_c$ (dashed, right axis) for several values of the treatment dose $q$. The dash-dotted line represents the time at which the treatment starts.

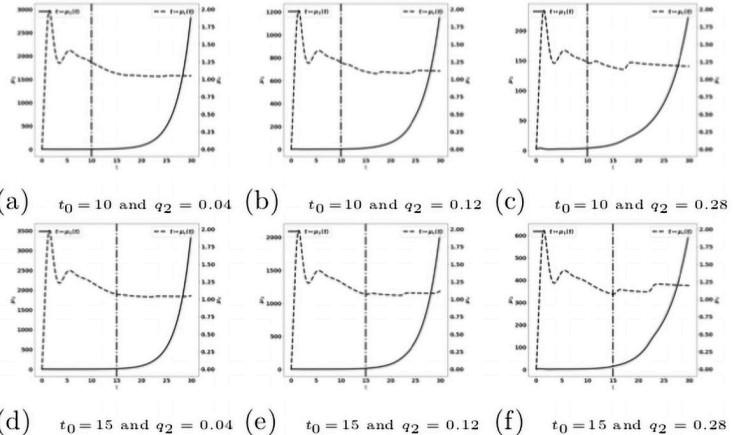

**Fig 21. Reduction of the protumor recruitment: Late administration of the treatment.** Simulation performed on the system (35a)–(35g), with (32)–(34). Evolution of the tumor mass $\mu_1$ (red curves, left axis), and of the immune strength $\bar{\mu}_c$ (blue curve, right axis) for several values of the treatment dose $q$. The dash-dotted line represents the time at which the treatment starts.

more efficient than the mono-therapies. Indeed a suitable combination of the treatment doses is able to control the tumor growth. For instance, the treatment based on the reactivation of exhausted immune cells fails in controlling the tumor when given at $t_0 = 10$ with a dose $q = 2$ see Fig 19(b), and the treatment based on cytokine/chemokine blockade fails with a dose $q_2 = 0.12$ at $t_0 = 10$, see Fig 21. However, the combination of the two treatments controls the tumor. Again, we observe that giving the treatments later requires to readjust the doses in order to control the tumor growth, see Fig 22. We notice that the controlled state contains residual tumor cells and activated immune cells, see Fig 23(b) and 23(c).

## Conclusion

This work introduces a mathematical model describing the interactions between tumor cells and the immune system that regulate tumor growth, taking into account the antagonistic effects of antitumor and protumor immune cells. While the antitumor action aims at eliminating tumor cells, the protumor effects favor its growth. The later can take different forms: elimination of antitumor cells, conversion of antitumor cells into protumor cells, or enhancement

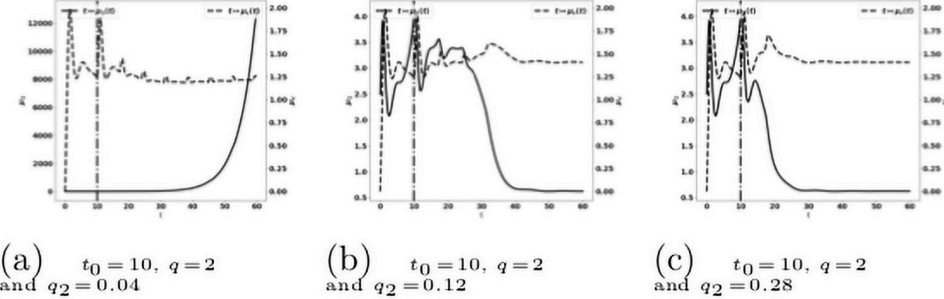

**Fig 22. Administration of the combined treatments at $t_0 = 15$.** Evolution of the tumor mass $\mu_1$ (plain, left axis), and of the immune strength $\bar{\mu}_c$ (dashed, right axis) for several values of the treatment dose $q$. The dash-dotted line represents the time at which the treatment starts.

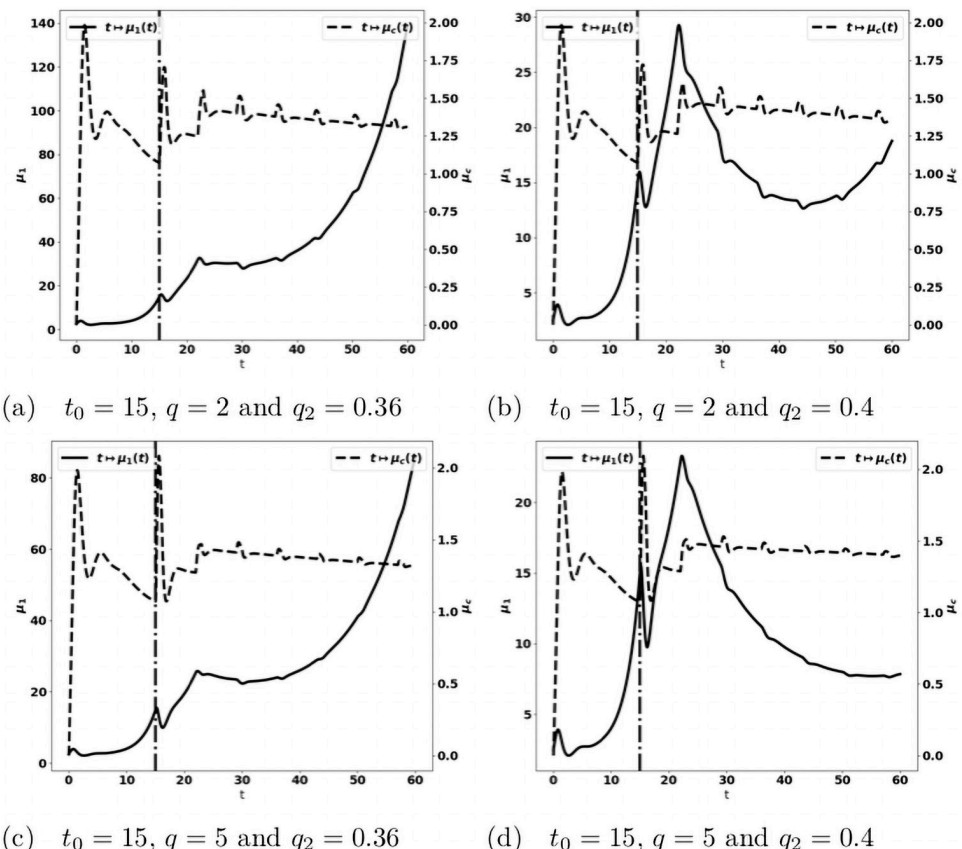

(a) $t_0 = 15$, $q = 2$ and $q_2 = 0.36$

(b) $t_0 = 15$, $q = 2$ and $q_2 = 0.4$

(c) $t_0 = 15$, $q = 5$ and $q_2 = 0.36$

(d) $t_0 = 15$, $q = 5$ and $q_2 = 0.4$

**Fig 23. Administration of the combined treatments at $t_0 = 10$.** Evolution of the tumor mass $\mu_1$ (plain, left axis), and of the immune strength $\bar{\mu}_c$ (dashed, right axis) for several values of the treatment dose $q$. The dash-dotted line represents the time at which the treatment starts.

of tumor growth. Mechanisms that dictate the balance between these two conflicting functions with the TME are still not clear, mainly because of their complexity and heterogeneity. Mathematical modeling can help capture such complexity. Our model based on partial differential equations is remarkably able to reproduce equilibrium and escape phases, depending on the value of the biological parameters.

Compared to the situation free of protumor activities we previously modeled [24], where the equilibrium seems to always occur (possibly on very long scale of time, though), the addition of protumor immune cells leads to uncontrolled tumor growth when tumor aggressiveness overcomes the efficiency of the antitumor immune responses. Our model thus pinpoints the critical role of the protumor immune response in the establishment of the escape phase. This is consistent with experimental and clinical data. Indeed, as reported in [41] the blockade of protumor neutrophil recruitment by cMet inhibitor decreases tumor growth and potentiates anti-PD1 immunotherapy. Similarly, anti-CD25 depleting antibody optimized to deplete Treg within tumors enhances anti-tumor immune responses and synergizes with anti-PD1 treatment [45]. Based on these findings, we used our model to study the effects on tumor progression of two common cancer therapeutic strategies, either the reactivation of hyporesponsive antitumor cells or the reduction of the recruitment of protumor cells. Such therapies boost the immune response and restore the equilibrium that maintains

the tumor in a viable state. Importantly, the numerical investigation brings out the influence of the starting time of the treatment and of the administrated dose. We also show on numerical grounds that combining the two approaches clearly improves the efficacy of the treatment. Such information is highly valuable and together with clinical observations, can comfort clinical decisions.

This preliminary study opens challenging perspectives. First, mathematical analysis can provide useful information, starting with further simplified equations, to understand the driving mechanisms of the equilibrium/escape phenomena and the effect of treatments. In particular this raises the practical issue of defining criteria that characterize the efficiency of the immune response, in line with RECIST recommendations [46]. Indeed, not only the residual mass of the tumor can be used as a relevant criterion, as in [36], but one has also to consider the time necessary to reach an equilibrium, as well as the features of the transient states. For treatments, the analysis, that should additionally consider toxicity effects, can help in understanding the optimal balance between dose and time of administration. Second, as simplified as it is, the model contains many parameters. Most of them are not known or even not easily accessible to experiments. Hence, based on optimisation techniques, an important work of parameter calibration should be performed from clinical data, with the two-fold difficulty that available data are rarely structured in time and space, and that data fitting techniques are far less developed for PDE than for ODE. Nevertheless, new mass cytometry imagery techniques open perspectives to address this issue [47]. Such investigation will permit us to determine relevant ranges for the parameters, which, in turn, will allow us to perform a detailed sensitivity analysis, beyond the attempt in [36]. This will be a decisive step to address in details the effects and the optimization of targeted treatments.

## Acknowledgments

The authors acknowledge the support of UCAncer, an incentive Université Côte d'Azur network, which has permitted and encouraged this collaboration.

## Author Contributions

**Conceptualization:** Kevin Atsou, Fabienne Anjuère, Véronique M. Braud, Thierry Goudon.

**Formal analysis:** Kevin Atsou, Véronique M. Braud, Thierry Goudon.

**Funding acquisition:** Fabienne Anjuère, Thierry Goudon.

**Investigation:** Kevin Atsou, Véronique M. Braud, Thierry Goudon.

**Methodology:** Kevin Atsou, Thierry Goudon.

**Software:** Kevin Atsou, Thierry Goudon.

**Supervision:** Véronique M. Braud, Thierry Goudon.

**Validation:** Kevin Atsou, Thierry Goudon.

**Visualization:** Kevin Atsou, Véronique M. Braud, Thierry Goudon.

**Writing – original draft:** Kevin Atsou, Fabienne Anjuère, Véronique M. Braud, Thierry Goudon.

**Writing – review & editing:** Kevin Atsou, Fabienne Anjuère, Véronique M. Braud, Thierry Goudon.

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
