## [Decision Letter · Decision Letter 0]

31 Aug 2021

PONE-D-21-19069

A size and space structured model of tumor growth describes a key role for protumor immune cells in breaking equilibrium states in tumorigenesis

PLOS ONE

Dear Dr. Goudon,

Thank you for submitting your manuscript to PLOS ONE. We do apologize for the delay with which we are returning you the referee comments. Unfortunately, one of the referees had some health problems and had to delay his report.

After careful consideration, we feel that it has merit but does not fully meet PLOS ONE’s publication criteria as it currently stands. Therefore, we invite you to submit a revised version of the manuscript that addresses the points raised during the review process.

As suggested by one of the referees, I would strongly invite you to revise your manuscript along the referee comments to improve the readability of your manuscript and to reach the criteria for a publication in PLOS ONE.

We look forward to receiving your revised manuscript.

Kind regards,

Christophe Letellier, Ph.D., Prof.

Academic Editor

PLOS ONE

Journal Requirements:

3. Thank you for stating the following in the Acknowledgments/Funding Section of your manuscript: 

This work was supported by the French Government (National Research Agency, ANR)

through the “Investments for the Future” programs LABEX SIGNALIFE ANR-11-

LABX-0028 and IDEX UCAJedi ANR-15-IDEX-01

This work was supported by the French Government (National Research Agency, ANR) through the “Investments for the Future” programs LABEX SIGNALIFE ANR-11- LABX-0028 and IDEX UCAJedi ANR-15-IDEX-01. The funders had no role in study design, data collection and analysis, decision to publish, or preparation of the manuscript.

Additional Editor Comments:

First, we do apologize the delay with which the referee comments are returned to you but one of the referees had some health problems and had to delay his report.

Second, both referees understood the potential of your work but they pointed out some problems in the presentation, inviting you to restructure your manuscript. I would like to invite you to follow this recommendations to improve the readability of your interesting work and thus, to have the positive feedback from the community it deserves.

Reviewers' comments:

Reviewer's Responses to Questions

**Comments to the Author**

1. Is the manuscript technically sound, and do the data support the conclusions?

Reviewer #1: Yes

Reviewer #2: Partly

2. Has the statistical analysis been performed appropriately and rigorously? 

Reviewer #1: N/A

Reviewer #2: N/A

3. Have the authors made all data underlying the findings in their manuscript fully available?

Reviewer #1: Yes

Reviewer #2: No

4. Is the manuscript presented in an intelligible fashion and written in standard English?

Reviewer #1: Yes

Reviewer #2: No

5. Review Comments to the Author

Reviewer #1: The authors are interested in immunoediting in tumors and in particular in the escape phase corresponding to an imbalance between an anti-tumor response and a pro-tumor response. The dynamics of the tumor are entirely driven by the immune system. The phenomena of angiogenesis and interactions with other healthy cells are ignored.

The proposed results are difficult to assess but appear biologically probable.

This work makes it possible to understand the failure of certain immunotherapies.

The fact that the solution to the cancer dynamic is based on the immune system does not allow us to understand the complexity of this pathology. It would have been interesting to combine this model with, for example, the process of angiogenesis and thus to test therapeutic combinations that already exist on the market.

Reviewer #2: The authors introduce a complex PDE model for anti-tumour/pro-tumour immune responses to cancer. Then, they consider a simplified version of this model (described by some simple ODEs) and for this model they investigate the existence and stability of various steady states. After that, the authors focus (I assume…) on the full model and try to find the equilibrium states. Then, the authors perform various numerical simulations showing what happens with model dynamics (although it is not clear whether they focus on the PDE or the ODE models) when various treatments are incorporated into the model.

Overall, the results have the potential to be very interesting. But the way they are presented makes it difficult to follow the flow of the paper. The authors should re-structure the paper and add more details, so that they “lead” the reader through their paper.

Below are some of the issues identified (which need to be addressed):

On page 11 the authors consider the case where “V and a are constant”. How is this possible when on the previous pages the authors show that “V” and “a” depend on “z” (see equations (1), (2) and (3))?

Page 14: I assume that the “healthy state (H)” is the one given by the zero vector on page 12? This needs to be made clear, since the notation (H) is never used on pages 12 and 13 ; only on page 14. Same comment about the other states: (NP) and (P).

Figure 3: It is not clear what is with the “expected equilibrium value a/\\delta (dotted)”. The tumour mass seems to approach this value at t=50 (panel (a)), t=10 (panel(b)), t=2.5 (panel (c)) and t=1.5 (panel(d)). The authors need to show on these panels what happens for very large time. Cutting the figures exactly at the time point where the tumour mass approaches this dotted line does not explain what is going on in Figure 3. Please re-do figure 3 to show also tumour dynamics for large time “t”.

The authors should also emphasise in the caption of this figure that there are two y-axes.

Finally, why is the continuous curve t-> mu1(t) but the dashed curve is t-> c (and not c(t))?

Figure 4 (c)-(d): I cannot see the dotted line. I can see only a dashed line…

Figure 4 (e )-(f): why the dashed curve is t->c while the continuous-dot curve is t->cr(t)? Why don’t you have t->c(t)?

Page 18: I don’t understand how the new section “Existence of equilibrium phases” is connected to the previous section. Does the new section correspond to the full model? How are the results in this Section different from the results in [24], [35], … Please explain very clearly what you do here. The reader cannot be let to “guess” the results of the section/paper, and how they connect with each other.

Figure 6: what represent the x-axis and y-axis in this figure? Same question for Figure 7.

Page 22: is “Emergent qualitative features …” a sub-section of the section “Results”? Or a parallel section? As discussed above, there is no explanation/flow for how the results connect with each other, so that the reader can follow easily this manuscript.

Figures 8,9,10, …14: Should I assume that these results are obtained with the PDE model, and the curves show space-averaged concentration values? Or do you show some simulations with an ODE model? Again, please don’t expect the reader to “guess” what you do here. You need to explain in detail what you show in these Figures.

Can the authors show also some space-time snapshots corresponding to the most interesting behaviours seen in Figures 8-10? (if these figures actually show spatially-averaged cell concentrations …)

Figures 16, 17, …: Same question as before: do the curves show space-averaged concentrations of cells (i.e., tumour mass and immune response)?

Are the Sections on pages 31, 34, 36 actually sub-sections of the “Results” section? It would be easier if sections/sub-sections/sub-subsections would be labelled.

6. PLOS authors have the option to publish the peer review history of their article (what does this mean?). If published, this will include your full peer review and any attached files.

Reviewer #1: No

Reviewer #2: No

---

## [Author Response · Author response to Decision Letter 0]

25 Sep 2021

We have revised the paper according to the reviewers and editors comments, paying attention to improve the presentation and readability and to make clear the organization. Detailed answers can be found in the files attached to the submission.

---

## [Decision Letter · Decision Letter 1]

18 Oct 2021

A size and space structured model of tumor growth describes a key role for protumor immune cells in breaking equilibrium states in tumorigenesis

PONE-D-21-19069R1

Dear Dr. Goudon,

We’re pleased to inform you that your manuscript has been judged scientifically suitable for publication and will be formally accepted for publication once it meets all outstanding technical requirements.

Kind regards,

Christophe Letellier, Ph.D., Prof.

Academic Editor

PLOS ONE

Additional Editor Comments (optional):

Thank you for having addressed the referee comments which are recommending now to publish your manuscript.

Reviewers' comments:

Reviewer's Responses to Questions

**Comments to the Author**

1. If the authors have adequately addressed your comments raised in a previous round of review and you feel that this manuscript is now acceptable for publication, you may indicate that here to bypass the “Comments to the Author” section, enter your conflict of interest statement in the “Confidential to Editor” section, and submit your "Accept" recommendation.

Reviewer #1: All comments have been addressed

Reviewer #2: (No Response)

2. Is the manuscript technically sound, and do the data support the conclusions?

Reviewer #1: Yes

Reviewer #2: (No Response)

3. Has the statistical analysis been performed appropriately and rigorously? 

Reviewer #1: Yes

Reviewer #2: (No Response)

4. Have the authors made all data underlying the findings in their manuscript fully available?

Reviewer #1: Yes

Reviewer #2: (No Response)

5. Is the manuscript presented in an intelligible fashion and written in standard English?

Reviewer #1: Yes

Reviewer #2: (No Response)

6. Review Comments to the Author

Reviewer #1: The authors have provided the necessary answers and clarifications to their work. The model is able to explain the dynamics of the complex interaction between the immune system and the tumour. However, I understand the need to simplify the tumour models, but in the future we cannot avoid introducing a system that takes into account the host and the toxicity of such treatments, particularly in the last part dealing with treatment strategies. Indeed, whether it is for checkpoint inhibitors which suffer in clinical practice from side effects sometimes so important that they cause a break or a cessation or CAR-T cells causing systemic hyperinflammation, it will be necessary and valuable to introduce these toxicities observed in clinical practice in order to obtain an efficient individualised strategy.

Reviewer #2: (No Response)

7. PLOS authors have the option to publish the peer review history of their article (what does this mean?). If published, this will include your full peer review and any attached files.

Reviewer #1: No

Reviewer #2: No

---

## [Editor Report · Acceptance letter]

9 Nov 2021

PONE-D-21-19069R1 

A size and space structured model of tumor growth describes a key role for protumor immune cells in breaking equilibrium states in tumorigenesis 

Dear Dr. Goudon:

I'm pleased to inform you that your manuscript has been deemed suitable for publication in PLOS ONE. Congratulations! Your manuscript is now with our production department. 

Kind regards, 

on behalf of

Professor Christophe Letellier 

Academic Editor

PLOS ONE